# Knowledge Crosswords: Geometric Reasoning over Structured Knowledge with Large Language Models

## Abstract

Large language models (LLMs) are widely adopted in knowledge-intensive tasks and have achieved impressive performance thanks to their knowledge abilities. While LLMs have demonstrated outstanding performance on atomic or linear (multi-hop) QA tasks, whether they can reason in knowledge-rich scenarios with interweaving constraints remains an underexplored problem. In this work, we propose *geometric reasoning over structured knowledge*, where pieces of knowledge are connected in a graph structure and models need to fill in the missing information of this graph. Such geometric knowledge reasoning would require the ability to handle structured knowledge, reason with uncertainty, verify facts, and backtrack when an error occurs. We specifically propose Knowledge Crosswords, a multi-blank QA dataset where each problem consists of a natural language question representing the geometric constraints of an incomplete entity network, where LLMs are tasked with working out the missing entities while meeting all factual constraints. Knowledge Crosswords contains 2,101 individual problems, covering a wide array of knowledge domains and further divided into three difficulty levels. We conduct extensive experiments to evaluate existing LLM prompting approaches on the Knowledge Crosswords benchmark. We additionally propose two new approaches, Staged Prompting and Verify-All, to augment LLMs' ability to backtrack and verify structured constraints. Our results demonstrate that while baseline approaches perform well on easier problems but struggle with questions on the hard side, our proposed Verify-All outperforms other methods by a large margin and is more robust with hard problems. Further analysis reveals that LLMs' ability of geometric reasoning over structured knowledge is still far from robust or perfect, susceptible to confounders such as the order of options, certain structural patterns, assumption of existence of correct answer, and more.

## 1 Introduction

Large language models (LLMs) have demonstrated an impressive ability on knowledge-intensive tasks such as open-domain QA (Petroni et al., 2019), misinformation detection (Karimi & Tang, 2019), and fact-checking (Gao et al., 2023). To assess the knowledge abilities of LLMs, existing tasks and datasets mostly focus on atomic (e.g., open-domain QA) (Rajpurkar et al., 2016; Das et al., 2022) or linear (e.g., multi-hop QA) (Press et al., 2022) settings, probing LLMs' responses to simple or multiple concatenated facts where each reasoning step has a unique definite answer. However, knowledge is not always arranged in a simple linear manner: it often involves more complex structural information, forming an interweaving network that connects various entities and relations through multiple chains as illustrated in Figure 1.

Consequently, an underexplored yet crucial question arises: *Can LLMs extend beyond linear compositionality and aggregate information from multiple chains along with various knowledge constraints?* Specifically, when certain pieces of knowledge are missing, can LLMs successfully fill in the blanks based on existing constraints represented by other available information in the network? In this work, we evaluate how well models can aggregate information from the given constraints across a graph representing pieces of knowledge and figure out the blanks in this graph. We term

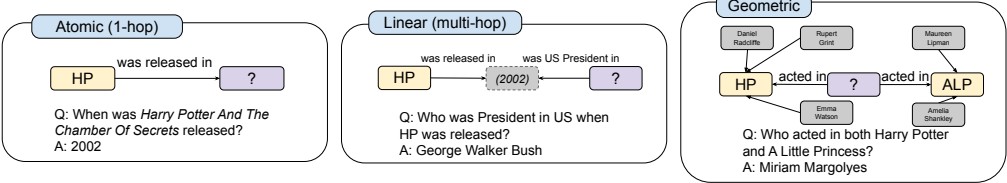

Figure 1: An illustration between the differences of atomic, linear, and geometric knowledge reasoning. Each reasoning step of atomic or linear QAs leads to a unique and definite (intermediate) answer, while multiple candidates exist before all constraints are jointly considered in geometric QA.

such ability *geometric reasoning over structured knowledge*. We believe that such reasoning introduces additional requirements for LLMs beyond those encountered in atomic or linear tasks, such as composing knowledge across multiple chains, reasoning with uncertainty, verification, backtracking, and more. To what extent could LLMs exhibit these abilities and handle contexts of geometric knowledge reasoning remain underexplored.

To this end, we propose KNOWLEDGE CROSSWORDS, a graph-based multi-blank QA dataset to evaluate whether LLMs could reason with structured knowledge bounded by geometric constraints. Each knowledge crossword consists of a list of constraints representing the edges of an incomplete graph of various entities, and the models need to reconstruct the graph by filling in blanks so that all constraints are met. To solve such problems, models need to consider candidates for each blank, jointly consider the provided constraints, verify whether to exclude or accept the candidates, and backtrack to pick different candidates when necessary until all constraints are met. We build individual knowledge crosswords by sampling subgraphs from an encyclopedic knowledge graph and randomly masking out certain entities as blanks. For each subgraph, we generate distractors of three difficulty levels for each blank to result in a multiple-choice setting. In total, KNOWLEDGE CROSSWORDS contains 2,101 problems, covering varying levels of difficulty, knowledge domains, and more.

We assess LLMs' performance on KNOWLEDGE CROSSWORDS with varying baseline approaches, ranging from simple zero-shot prompting to advanced ones such as self-consistency (Wang et al., 2022) and least-to-most prompting (Zhou et al., 2022). We find that all baseline approaches struggle with harder problems and have significant performance drops when no relevant knowledge is provided, while advanced prompting methods barely improve performance due to their reliance on left-to-right reasoning patterns. To address these challenges, we introduce two new instruction-based techniques, STAGED PROMPTING and VERIFY-ALL, aiming at augmenting LLMs' abilities for backtracking, constraint verification, and more. We find that VERIFY-ALL achieves top performance and is more robust with harder settings, while the success of STAGED PROMPTING is contingent on stronger base LLMs. Further analysis reveals that the geometric knowledge reasoning ability of LLMs is far from robust, as it can be susceptible to a variety of factors such as option order, "None of the above" possibilities, option amount, certain structural patterns, and more. We envision KNOWLEDGE CROSSWORDS as a comprehensive testbed to evaluate LLM knowledge abilities in more complex and structured settings.

## 2 THE KNOWLEDGE CROSSWORDS BENCHMARK

We propose the KNOWLEDGE CROSSWORDS benchmark, a graph-based multi-blank QA dataset to evaluate whether LLMs could reason with structured knowledge bounded by geometric constraints (Appendix C). An overview of knowledge crossword is illustrated in Figure 2.

**Definition** Each knowledge crossword consists of a question graph $\mathcal{G}_\mathcal{Q} = \{(h, r, t)|h, t \in \mathcal{V}_\mathcal{Q}, r \in \mathcal{R}\}$, where $\mathcal{V}_\mathcal{Q}$ is the set of entities represented as nodes of $\mathcal{G}_\mathcal{Q}$ and $\mathcal{R}$ is the set of all possible relations between entities. Each $(h, r, t)$ in $\mathcal{G}_\mathcal{Q}$ denotes a directed edge representing a factual association such as (Marvin Minsky, has won prize, Turing award). In the question graph $\mathcal{G}_\mathcal{Q}$, certain nodes $b_i \in \mathcal{V}_\mathcal{Q}$ are masked out as blanks $B = [b_1, b_2, \ldots, b_{|B|}]$. The goal of each knowledge crossword

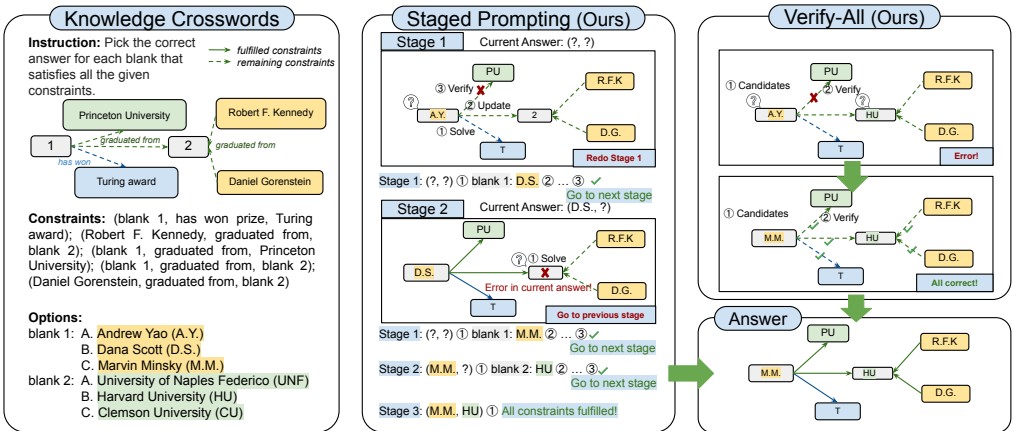

Figure 2: Overview of KNOWLEDGE CROSSWORDS and two proposed approaches, STAGED PROMPTING and VERIFY-ALL. Each individual knowledge crossword includes task instruction, description of the structured factual constraints, and multiple-choice QA options. In each stage of STAGED PROMPTING, LLMs ① *solve* a blank based on one remaining constraint (dashed edge); ② *update* the status by filling in the proposed answer; then ③ *verify* filled constraints to proceed or backtrack. In VERIFY-ALL, LLMs propose a combination of ① *candidates* and ② *verify* all constraints with those candidates, and repeat this process until all constraints are met.

is to find one combination of answers for all blanks $A = [a_1, a_2, \ldots, a_{|B|}]$ that satisfy all factual associations represented as geometric constraints in the question graph $\mathcal{G}_\mathcal{Q}$.

**Data Source** We resort to encyclopedic knowledge graphs, specifically YAGO (Suchanek et al., 2023), as scaffolds of geometric knowledge reasoning to construct the KNOWLEDGE CROSSWORDS benchmark. We conduct preprocessing to remove certain relations in the YAGO knowledge graph that are location-related, time-sensitive, or not self-evident. This is to ensure that the KNOWLEDGE CROSSWORDS is minimally affected by question ambiguity, outdated knowledge, etc. We obtain the filtered knowledge graph as $\mathcal{KG} = \{(h, r, t)|h \in \mathcal{H}, r \in \mathcal{R}, t \in \mathcal{T}\}$, where $\mathcal{H}$, $\mathcal{R}$, and $\mathcal{T}$ are the sets of heads, relations, and tails respectively. We proceed to use the filtered YAGO knowledge graph as the data source for the KNOWLEDGE CROSSWORDS benchmark.

**Generating Question Graphs** We first adopt two hyperparameters to control the property and difficulty of the generated question graphs: *question graph size $s_G$*, representing the total number of nodes in a question graph, and *blank size $s_B$*, representing the number of nodes masked out as blanks. To obtain question graphs, we start from a random center node $c$ and retrieve the $k$-hop neighborhood of $c$ as $\mathcal{G}_\mathcal{N}^{(c)}$. We then downsample $\mathcal{G}_\mathcal{N}^{(c)}$ by randomly removing nodes with higher degrees than a dynamic threshold $t_d$ in the original knowledge graph $\mathcal{KG}$, until the largest weakly connected component in $\mathcal{G}_\mathcal{N}^{(c)}$ has a size no greater than $s_G$. This is motivated by the fact that entities with higher degrees are presumably less typical and more ambiguous, potentially causing distraction in the model's reasoning (Shomer et al., 2023; Qian et al., 2023). We refer to the largest connected component in downsampled $\mathcal{G}_\mathcal{N}^{(c)}$ as answer graph $\mathcal{G}_\mathcal{A}$.

We then randomly select $s_B$ nodes in $\mathcal{G}_\mathcal{A}$ from those with higher degrees than another dynamic threshold $t_b$ in $\mathcal{G}_\mathcal{A}$ and mask them out as blanks $B$. These high-degree blanks would be rich in geometric associations and provide LLMs with more constraints to work with. We use $s_G$, $s_B$, as well as the degree of the center node $c$ and dynamic thresholds for the removing and masking steps $t_d$ and $t_b$ as difficulty control measures to diversify problems in the KNOWLEDGE CROSSWORDS benchmark. As the question graph generation step does not guarantee answer uniqueness, we exhaustively search answers for each $\mathcal{G}_\mathcal{Q}$ in $\mathcal{KG}$ and only retain those with one valid answer combination.

**Negative Sampling for Multiple-Choice QA** We mainly consider KNOWLEDGE CROSSWORDS under a multiple-choice QA setting, where several options are provided for each blank in $\mathcal{G}_\mathcal{Q}$. This would require negative sampling for each blank to provide distractors in addition to the correct answer, while we identify a taxonomy of three progressive rules for distractors from loose to strict:

- Rule #1: **Same semantic role as the blank.** If a blank $b_i$ is the head (or tail) of an edge with relation $r_i$, then the distractor for $b_i$ should be selected from the set of heads (or tails) of edges with the same relation $r_i$.

- Rule #2: **In the neighborhood of the blank.** The distractor for $b_i$ should occur in the neighborhood $\mathcal{G}_\mathcal{N}^{(c)}$ around the center node $c$ used to generate $\mathcal{G}_\mathcal{Q}$. This rule further narrows the range of such edges so that the distractors are more likely to be in a similar context as the blank.

- Rule #3: **Fulfill a definite constraint if any.** If the other end of the edge that the blank $b_i$ is incident to is known, then we say such edge is a definite constraint for $b_i$, and the distractor should satisfy at least one of such definite constraints for $b_i$ to fulfill Rule #3. Such distractors impose higher demands on language models in the sense that LMs should jointly consider all constraints to exclude them.

As a result, we obtain three negative sampling strategies with varying difficulty implications for knowledge crosswords: *easy*, where distractors only meet Rule #1; *medium*, where distractors meet Rule #1 and #2; *hard*, where distractors meet Rule #1, #2, and #3. We opt to separately assign multiple options for each blank in $\mathcal{G}_\mathcal{Q}$, using either *easy*, *medium*, or *hard* strategies. Most blanks have three options including the correct answer and there are blanks with only two available options.

**Relevant Knowledge** We additionally consider two variants regarding external knowledge availability: an **open-book** setting, where no external knowledge is provided and LLMs need to solve knowledge crosswords with only their internal parametric knowledge; a **closed-book** setting, where external knowledge, both useful and confounding, is provided in a preamble to the problem description. LLMs would need to dynamically select useful information to facilitate geometric knowledge reasoning. Concretely, we similarly sample confounding knowledge triples from $\mathcal{KG}$, guided by the three rules, to obtain confounding context for each $\mathcal{G}_\mathcal{Q}$. Useful and confounding knowledge are then downsampled to 1:3, shuffled, and presented before each knowledge crossword. In this way, we provide both necessary knowledge and confounding information, making the solving process non-trivial.

**Evaluation Metrics** We evaluate performance on KNOWLEDGE CROSSWORDS with two metrics: *Partial-Credit* (PC), indicating the portion of blanks that have been answered correctly; *Full-Credit* (FC), indicating whether all blanks are correct in a given knowledge crossword. Formally,

$$\text{PC} = \frac{\sum_{i=1}^{s_B} \mathbf{1}[A_i^* = A_i]}{s_B}, \quad \text{FC} = \mathbf{1}[\text{PC} = 1],$$

where $A_i^*$ denotes the prediction of $B_i$ by LLMs and $\mathbf{1}[]$ denotes the indicator function.

**Benchmark Statistics** After the previous steps, we obtain 873 valid question graphs with different levels of scales and sparsity. Each question graph is then used to construct three problems using the three levels of negative sampling difficulty, *easy*, *medium*, and *hard*, enabling fine-grained evaluation. The benchmark statistics are shown in Table 1.

| Subset | #Qs | Avg. #Nodes | Avg. #Edges | Avg. #Blanks |
|---|---|---|---|---|
| EASY | 869 | 7.28 | 6.63 | 2.89 |
| MEDIUM | 873 | 7.28 | 6.64 | 2.89 |
| HARD | 359 | 7.09 | 6.41 | 2.62 |

Table 1: Statistics of the KNOWLEDGE CROSSWORDS Benchmark. We report the number of questions and the average number of nodes, edges, and blanks for each subset with different negative sampling difficulty.

## 3 OUR APPROACH

We hypothesize that the left-to-right reasoning patterns in autoregressive language models (Yao et al., 2023) and prompting approaches would fall short of solving knowledge crosswords, which require backtracking, maintaining problem states, verifying existing information, reasoning with structured constraints, and more. To this end, we introduce two instruction-based methods that promote these abilities, illustrated with a detailed example in Figure 2.

### 3.1 STAGED PROMPTING

The STAGED PROMPTING approach divides geometric knowledge reasoning into stages involving three steps: *solve*, *update*, and *verify*. At the beginning of each stage, LLMs maintain a current status

| Method | w/ Relevant Knowledge | | | | | | w/o Relevant Knowledge | | | | | |
|---|---|---|---|---|---|---|---|---|---|---|---|---|
| | easy | | medium | | hard | | easy | | medium | | hard | |
| | PC | FC | PC | FC | PC | FC | PC | FC | PC | FC | PC | FC |
| RANDOM | 34.3 | 6.1 | 34.2 | 5.5 | 33.5 | 8.4 | 34.3 | 6.1 | 34.2 | 5.5 | 33.5 | 8.4 |
| UPPERBOUND | 98.8 | 96.7 | 99.1 | 97.4 | 91.8 | 82.2 | - | - | - | - | - | - |
| ZERO-SHOT | 97.3 | 93.7 | 97.4 | 94.2 | 77.9 | 55.4 | 81.3 | 57.1 | 83.3 | 60.6 | 57.2 | 24.8 |
| FEW-SHOT | 97.8 | 93.2 | 97.6 | 93.5 | 78.8 | 54.0 | 83.7 | 60.8 | 84.7 | 63.3 | 56.8 | 25.3 |
| CoT | 94.6 | 86.5 | 95.0 | 88.9 | 77.9 | 56.3 | 74.0 | 44.0 | 76.4 | 48.5 | 55.7 | 27.0 |
| CoT+SC | 95.9 | 89.8 | 96.6 | 91.2 | 78.7 | 57.4 | 75.2 | 45.8 | 77.3 | 49.1 | 56.7 | 28.4 |
| LtM | 86.0 | 68.9 | 86.3 | 68.6 | 69.6 | 43.5 | 75.6 | 47.3 | 76.6 | 48.2 | 51.1 | 19.2 |
| STAGED PROMPTING | 91.9 | 81.6 | 91.2 | 80.4 | 70.5 | 44.5 | 64.3 | 34.3 | 67.4 | 38.3 | 47.9 | 15.8 |
| VERIFY-ALL | **98.6** | **96.1** | **98.7** | **96.2** | **83.9** | **64.6** | **84.5** | **62.3** | **86.1** | **66.9** | **59.7** | **29.8** |

Table 2: Model performance with GPT-3.5-TURBO. PC indicates Partial-Credit and FC indicates Full-Credit. The best results are **bold-faced**, and the second-best ones are underlined. Detailed prompts can be found in Appendix E.

| Method | w/ Relevant Knowledge | | | | | | w/o Relevant Knowledge | | | | | |
|---|---|---|---|---|---|---|---|---|---|---|---|---|
| | easy | | medium | | hard | | easy | | medium | | hard | |
| | PC | FC | PC | FC | PC | FC | PC | FC | PC | FC | PC | FC |
| RANDOM | 34.3 | 6.1 | 34.2 | 5.5 | 33.5 | 8.4 | 34.3 | 6.1 | 34.2 | 5.5 | 33.5 | 8.4 |
| *Experiments with GPT-4* | | | | | | | | | | | | |
| STAGED PROMPTING | **99.1** | **98.8** | **96.3** | 95.6 | **95.4** | **94.2** | 75.4 | 70.7 | 78.8 | 74.0 | 52.3 | 32.4 |
| VERIFY-ALL | 98.1 | 98.1 | 95.7 | **95.7** | 92.8 | 90.5 | **88.0** | **83.4** | **89.5** | **85.5** | **59.5** | **38.6** |

Table 3: Model performance (%) with different methods using GPT-4. PC indicates Partial-Credit and FC indicates Full-Credit. The best results are **bold-faced**. With relevant knowledge prepended, STAGED PROMPTING generally outperforms VERIFY-ALL, and VERIFY-ALL outperforms STAGED PROMPTING when no relevant knowledge is provided. Note that the easy and medium subsets are slightly downsampled in several settings due to computational costs.

of solved blanks and unresolved constraints (edges that involve unsolved blanks). In the *solve* step, LLMs propose a candidate for an unsolved blank based on either internal or external knowledge; in the *update* step, LLMs update unsolved constraints using the newly proposed candidate for a given blank; in the *verify* step, LLMs reflect on the updated constraints in the *update* step and judge their validity. If an invalid factual association is identified as a result of the proposed candidate, LLMs backtrack to the problem status in the previous stages and propose another option; otherwise, LLMs proceed to tackle the remaining blanks until all blanks are filled and all constraints are met.

## 3.2 VERIFY-ALL

While STAGED PROMPTING presents an elaborate problem-solving process that tackles challenges such as backtracking and status updates, such complex reasoning might be hard to learn in context for LLMs. We additionally propose the VERIFY-ALL approach: candidates for each blank are simultaneously proposed, rather than in separate stages. A verification step is then employed to assess the validity of all filled constraints using these proposed candidates. If an error is detected, the LM should backtrack and propose an alternative set of candidates until no error is found.

## 4 EXPERIMENT SETTINGS

### 4.1 BASELINES

We adopt various prompting techniques as baselines, including zero-shot (ZERO-SHOT) prompting, few-shot in-context learning (FEW-SHOT), Chain-of-Thought prompting (CoT), CoT with self-consistency (CoT+SC), and least-to-most prompting (LtM). Besides, we adopt the RANDOM baseline which refers to randomly selecting an option for each blank. We also present an UPPERBOUND baseline, where we present the constraints in $G_Q$ filled with correct answers as relevant knowledge.

| w/ NOTA? | w/ correct options? | easy | | medium | | hard | |
|:---:|:---:|:---:|:---:|:---:|:---:|:---:|:---:|
| | | PC | FC | PC | FC | PC | FC |
| ✓ | ✗ | 35.7 | 14.0 | 35.6 | 13.5 | 11.4 | 3.1 |
| ✓ | ✓ | 68.1 | 46.8 | 69.1 | 48.3 | 50.7 | 20.6 |
| ✗ | ✓ | **83.7** | **60.8** | **84.7** | **63.3** | **56.8** | **25.3** |

Table 4: Model performance (%) evaluated with FEW-SHOT under the w/o Relevant Knowledge setting. The best results are **bold-faced** and the second-best ones are underlined. "NOTA" denotes the instruction "Output 'None of the above' if none of the option combinations satisfy all the constraints."; "correct options" denotes that the correct options are provided as a candidate option and LLMs should output the answer instead of "None of the above" when correct options exist. Specifically, row 1 and row 2 correspond to setting #1 and #2 respectively.

## 4.2 MODELS AND SETTINGS

Unless otherwise specified, we use ChatGPT (GPT-3.5-TURBO) as the base language model in our experiments, and we additionally test out GPT-4 with our approaches as reported in Table 3 and the open-source Llama 2 (Touvron et al., 2023) in Section 6. For few-shot prompting techniques (FEW-SHOT, CoT, CoT+SC, LTM), we present 5 in-context exemplars. The sampling temperature $\tau$ is set to $0.1$ except for self-consistency; we sample 5 Chain-of-Thought responses with temperature $\tau = 0.7$ for the self-consistency baselines.

## 5 RESULTS

We evaluate various approaches on KNOWLEDGE CROSSWORDS and present results in Table 2.

**LLMs have preliminary abilities for geometric knowledge reasoning.** Table 2 shows that all investigated approaches outperform RANDOM baseline by a large margin despite the difficulty of the task, while LLMs could achieve a 90+ Full-Credit score on simple problems and settings (e.g., easy and w/ knowledge).

**All approaches exhibit severe performance drops on harder knowledge crosswords.** While it might be relatively straightforward for LLMs to eliminate false options that satisfy few definite constraints in the knowledge crosswords, they face difficulties in excluding options that meet a subset of all the constraints presented. This indicates that LLMs' abilities for geometric knowledge reasoning are greatly impacted by how confounding distractors are, meaning that LLMs are far from robust in complex structured knowledge contexts.

**Relevant knowledge does help LLMs solve knowledge crosswords.** Despite the existence of confounding information, LLMs do benefit from the relevant knowledge provided. On average, the w/ knowledge setting exhibits a 34.3% FC gain compared to w/o knowledge settings. However, it remains unclear whether LLMs obtain such performance gain by effectively understanding and leveraging the relevant knowledge or by spurious correlations between relevant knowledge and given constraints.

**Existing advanced prompting methods show little improvement.** Specifically, CoT, LTM and CoT+SC have little improvement compared to basic ZERO-SHOT and FEW-SHOT prompting. The average PC and FC of CoT, LTM and CoT+SC is 5.6% and 10.4% **less** than those of ZERO-SHOT and FEW-SHOT. This suggests that the left-to-right reasoning patterns employed by these prompting techniques may not be suitable for knowledge crosswords, as these prompting methods fail to effectively encourage LLMs to perform the non-progressive steps of verification and backtracking when necessary.

**Adding verification and backtracking steps improves geometric reasoning ability.** With GPT-3.5-TURBO, a simple verification step in the instruction brings a huge gain for VERIFY-ALL. Interestingly, after a closer look into the responses from GPT-3.5-TURBO, we find that only a very limited number of responses involve detecting errors in proposed candidates and backtracking accordingly. This indicates that the performance gain mainly comes from LLMs proposing more precise answers in a single attempt with the instruction that they need further verification.

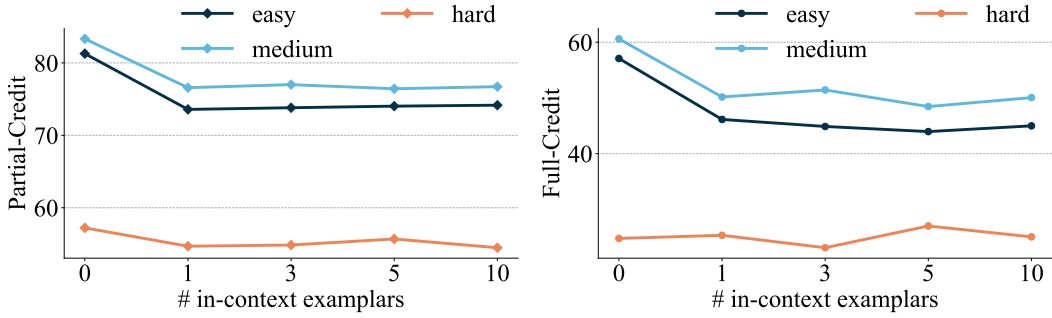

Figure 3: Model performance (%) using CᴏT under the w/o Relevant Knowledge setting with an increasing number of in-context exemplars. An increase in the number of exemplars does not bring performance gain.

Furthermore, while GPT-3.5-TURBO struggles at learning complex reasoning steps demonstrated in STAGED PROMPTING, Table 3 shows that STAGED PROMPTING achieves impressive results with GPT-4 and generally outperforms all other methods including VERIFY-ALL when relevant knowledge are provided. This indicates that the more elaborate instructions of STAGED PROMPTING work best with more advanced LLMs, as smaller models struggle to grasp these detailed reasoning steps.

# 6 ANALYSIS

**None of the above** In the main experiments, the correct answer is always provided as one of the options for each blank and LLMs are instructed to pick the correct answer. To study whether LLMs would be robust if we tell LLMs that there could be such cases where none of the options is correct, we add additional instruction *"Output 'None of the above' if none of the option combinations satisfy all the constraints."* and evaluate the performance of GPT-3.5-TURBO with the FEW-SHOT prompting. Specifically, we experiment with two settings: #1. The correct option is removed from candidates and LLMs should output 'None of the above'; #2. The correct option occurs in the candidates despite the additional instruction and LLMs should output the unique correct answer for each blank. Table 4 demonstrates similar results as shown by Kadavath et al. (2022), GPT-3.5-TURBO seldom outputs "None of the above" even if the correct answer is not provided as one option (row 1 & row 2), while it does get affected by the absence of the assumption that correct option always exists (row 2 & row 3).

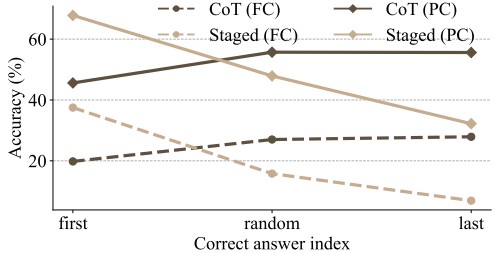

Figure 4: Model performance (%) under the w/o Relevant Knowledge setting using CᴏT and STAGED PROMPTING with different orders of options evaluated on the hard subset.

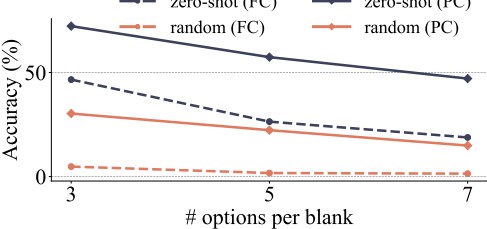

Figure 5: Model performance (%) evaluated on 292 problems using ZERO-SHOT for increasing number of options per blank. RANDOM denotes the baseline of random guess.

**Order of options** As STAGED PROMPTING considers one candidate at one time, we expect the performance may be negatively correlated with the index of the correct answer in the list of options. Figure 4 demonstrates that the performance of STAGED PROMPTING declines when the correct answer appears later. Such a negative correlation could be a manifestation of LLM hallucination (Ji et al., 2023). On the other hand, we see an opposite trend in the performance of CᴏT. This

---

[0]Within 4k-context, in the w/o Relevant Knowledge setting, the numbers of finished responses for easy, medium, and hard questions are 755, 781, and 341 respectively; in the w/ Relevant Knowledge setting, the numbers of finished responses for easy, medium, hard questions are 759, 769 and 328 respectively. The credits are calculated based on these finished responses only.

indicates that CoT does not consider the options sequentially and later options may be paid more attention than earlier ones.

**Number of options** As the number of options for each blank increases, the problem becomes harder due to the presence of more confounders. We expect to see a downward trend in model performance when there are more distractors per blank. Unsurprisingly, the results in Figure 5 show that the performance is negatively correlated with the number of options per blank. We also observe that performance gap with random guessing is narrowing, suggesting that KNOWLEDGE CROSSWORDS might be difficult for LLMs in the open-book generation setting. (Appendix A)

**Number of in-context examplars** Despite the in-context learning ability demonstrated by LLMs (Brown et al., 2020), we find that more in-context exemplars fail to improve model performance on KNOWLEDGE CROSSWORDS. As presented in Figure 3, for questions with all three difficulty levels, the best performance is achieved at ZERO-SHOT except for the Full-Credit of hard problems. This indicates that left-to-right CoT reasoning could not be adequately learned in context for the problem of knowledge crosswords.

**Specific structural patterns** We observe that the occurrence of certain structural patterns in the question graph makes the problem harder for LLMs. We identify three special structural patterns: 1) *A-B*, where two blanks are connected by an edge; 2) *A-B-C*, where three blanks are linked together; 3) *cycle*, where blanks form a cycle. According to Figure 6, we see a decrease in the accuracy in the settings of *A-B* and *A-B-C*, compared to performance on the whole benchmark. We find performance gains in problems with *cycle*s, while such problems are very limited in KNOWLEDGE CROSSWORDS.

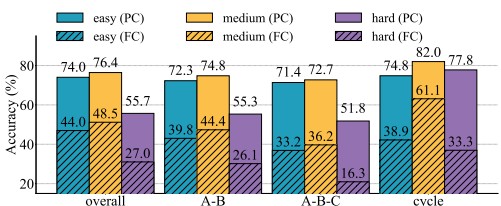

Figure 6: Model performance (%) for problems with specific structural patterns. "overall" denotes the performance calculated over the whole benchmark.

**Difficulty of in-context examplars** We investigate the correlation between the difficulty of in-context exemplars and model performance by evaluating the performance with 5-shot CoT using 4 different sets of in-context exemplars. Specifically, we use four sets of in-context exemplars: *easy*, where all examples come from the easy subset of KNOWLEDGE CROSSWORDS; similarly *medium* and *hard*; *mixed*, where a mixture of 2 easy, 2 medium, and 1 hard examples are incorporated. Table 5 demonstrates that for problems of different difficulties, LLMs learn better from medium or mixed in-context exemplars.

| examplar Q | test Q | | | | | |
| | easy | | medium | | hard | |
| | PC | FC | PC | FC | PC | FC |
| --- | --- | --- | --- | --- | --- | --- |
| EASY | 73.9 | 44.4 | 75.9 | 47.9 | 55.2 | 24.0 |
| MEDIUM | **75.4** | **47.9** | **77.2** | **49.7** | 55.1 | 25.9 |
| HARD | 73.4 | 43.5 | 76.2 | 48.7 | 55.3 | 24.0 |
| MIXED | 74.9 | 46.0 | 75.7 | 47.4 | **55.8** | **26.5** |

Table 5: Model performance (%) with CoT using exemplars of different difficulties under the w/o Relevant Knowledge setting. The best results are **bold-faced**.

This indicates that LLMs best conduct geometric knowledge reasoning when problems and solutions with a range of difficulty levels are presented, enabling progressive in-context learning from simple to hard. As a result, we use in-context exemplars with mixed difficulties in the experiments.

**Fine-tuning and open-source LMs** We additionally evaluate the geometric knowledge reasoning abilities of an open-source language model - LLAMA2-7B with 100 problems randomly selected across all difficulty subsets. Without fine-tuning, LLAMA2-7B demonstrates a performance close to random guess. After instruction-tuning with 1,471 knowledge crosswords randomly selected from all 2,101 questions, the Partial-Credit and Full-Credit become 17.7% and 12.0% higher than ZERO-

| Method | PC | FC |
| --- | --- | --- |
| RANDOM | 32.8 | 5.0 |
| ZERO-SHOT | 29.6 | 5.0 |
| FEW-SHOT | 35.5 | 7.0 |
| INSTRUCTION-TUNING | 47.3 | 17.0 |

Table 6: Performance (%) of LLAMA2-7B on a dataset subset.

SHOT prompting as reported in Table 6. This indicates that instruction tuning (Wei et al., 2021) could augment LLMs for solving knowledge crosswords, while to what extent they work with larger LLMs requires further research.

# 7 RELATED WORK

**Understanding and Expanding the Knowledge Abilities of LLMs** After trained on massive textual corpora, LLMs tend to encode a substantial amount of factual knowledge in their parametric memory (Mallen et al., 2023). Since the advent of works like LAMA (Petroni et al., 2019), numerous studies have sought to investigate the extent to which LLMs encode and retrieve factual knowledge (Yu et al., 2022) and to understand the models' capacity to utilize this parametric knowledge effectively (Mallen et al., 2023). Mruthyunjaya et al. (2023) show that while LLMs possess the potential to recall factual information, their ability to capture complex topological and semantic traits of KGs remains notably limited. In addition, LLMs encounter challenges related to knowledge update (Hase et al., 2023), knowledge conflict (Chen et al., 2022), irrelevant context (Shi et al., 2023a), long-tail knowledge (Kandpal et al., 2023; Sun et al., 2023) and more.

As a result, various methods have been proposed to further expand the knowledge abilities of language models, including prompting techniques (Press et al., 2022; Sun et al., 2022; Yu et al., 2022; Kojima et al., 2022; Ye & Durrett, 2022), retrieval augmentation (Shi et al., 2023b), search engine integration (Yu et al., 2022; Press et al., 2022), multi-LLM collaboration (Feng et al., 2023), and more. While these works primarily focus on evaluating and improving abilities to handle atomic (e.g., open-domain QA) or linear (e.g., multi-hop QA) knowledge, we propose to assess whether LLMs could reason with structured knowledge bounded by geometric constraints that are better aligned with the structural nature of knowledge.

**Reasoning with Large Language Models** LLMs have been evaluated on a myriad of reasoning tasks in an in-context learning setting, including math problems (Ling et al., 2017; Lewkowycz et al., 2022), logical reasoning (Srivastava et al., 2023; Huang et al., 2022), factual knowledge reasoning (Press et al., 2022), commonsense reasoning (Talmor et al., 2019), and more. Multiple prompting techniques have been developed to promote the reasoning ability of LLMs. Leveraging the in-context learning behavior of LLMs, various prompting techniques (Wei et al., 2022; Zhou et al., 2022; Khot et al., 2022; Wang et al., 2022) have been proposed to boost the reasoning ability. Specifically, Khot et al. (2022) and Yao et al. (2023) incorporate programs as guides to LLM generation. In our work, we mainly focus on evaluating the LLM itself for its geometric reasoning ability, with minimal guidance from explicit code or program-based guidance. And the prompting techniques introduced in previous work barely improve the performance in KNOWLEDGE CROSSWORDS as their inherent left-to-right reasoning patterns are not perfect for our problems. Experiments by Kadavath et al. (2022) demonstrate that self-evaluation improves with model size, which in effect means that verification has an advantage over generation. Following this finding and considering the nature of our problems, we harness the self-evaluation ability of LLMs to improve the overall accuracy of their responses.

# 8 CONCLUSION

In this work, we propose KNOWLEDGE CROSSWORDS, a multi-blank QA dataset where each problem consists of a natural language question representing the geometric constraints of an incomplete entity network and LLMs are tasked with working out the missing entities while meeting all factual constraints. We envision KNOWLEDGE CROSSWORDS as a comprehensive testbed to evaluate whether LLMs could perform geometric reasoning over structured knowledge. We adopt a variety of prompting methods on KNOWLEDGE CROSSWORDS and find that all investigated approaches perform above random, while we further propose STAGED PROMPTING and VERIFY-ALL to tackle unique challenges in geometric knowledge reasoning such as backtracking and fact verification. As a result, VERIFY-ALL outperforms all other techniques with ChatGPT while STAGED PROMPTING works better with the more advanced GPT-4. Further analysis shows that the geometric reasoning ability of LLMs over structured knowledge is far from robust, as it is impacted by factors such as the order of options, "None of the above" scenarios, certain structure patterns entailing greater uncertainty, and more.

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

## A DISCUSSION

**Geometric Reasoning in an open-book generation setting**    While we mainly focus on solving knowledge crosswords in a multiple-choice setting, it is interesting to evaluate the geometric reasoning ability in an open-book generation setting. Specifically, the problems in KNOWLEDGE CROSSWORDS have unique answers, which should be useful when switching to an open-book generation setting as answer uniqueness makes evaluation easier and makes the problem clearer. Our preliminary experiments show that solving knowledge crosswords in the open-book generation setting is much harder. Considering the model performance in the multiple-choice setting, one method that might be promising is to prompt LLMs themselves to generate candidates for each blank and thereby transform the open-book generation problem into a multiple-choice problem.

**Performance gain of VERIFY-ALL**    While VERIFY-ALL helps LLMs obtain large performance gains in solving knowledge crosswords, it is quite intriguing when investigating where such gains come from. Specifically, in the w/ Relevant Knowledge setting, among all 359 hard problems, we find only 3 problems whose solution with VERIFY-ALL involves detecting errors in verification and re-propose candidates. Among the 3 problems, 2 are answered correctly by both VERIFY-ALL and CoT, and both methods fail the other problem. This leads to an interesting implication that the performance gain comes from LLMs proposing more precise answers in the first attempt, and that LLMs can jointly consider all constraints rather than consider one by one. We envision the study of such performance gain and the application of the insight as possible future directions.

**Application of geometric reasoning over structured knowledge**    Despite the difficulty of the task, LLMs do show preliminary geometric reasoning ability over structured knowledge. While such ability still has a long way to achieve perfection, this finding opens up the possibility of using LLMs as flexible relational databases and accessing the parametric knowledge with prompts similar to SQL (structured query language).

**Same prompting approach with different LLMs**    While GPT-3.5-TURBO does not benefit from STAGED PROMPTING, experiments using STAGED PROMPTING with GPT-4 demonstrate impressive results under the w/Relevant Knowledge setting. Taking a close look at the responses of GPT-3.5-TURBO, we find they fail to follow the reasoning steps presented in the exemplars even if we facilitate the process by guiding the *update* step with program. On the other hand, GPT-4 learn better from exemplars of STAGED PROMPTING with similar settings. This indicates that the success of STAGED PROMPTING relies heavily on the choice of LLMs.

## B LIMITATIONS

**Limited data source**    We construct KNOWLEDGE CROSSWORDS based on only the encyclopedic knowledge graph YAGO, which covers topics on general knowledge about people, cities, countries, movies, and organizations from Wikidata. Since we will make the code publicly available, we leave it to future work on evaluating the geometric reasoning ability of LLMs on different topics with various data sources, such as biomedical knowledge graphs (Chang et al., 2020) and networks in social sciences (Feng et al., 2022).

**Answer Uniqueness**    Due to the incompleteness of knowledge graphs, it is possible that the answer to a problem in KNOWLEDGE CROSSWORDS is not unique. However, such likelihood is presumably low and does not hurt the general evaluation of the geometric reasoning ability of LLMs.

## C KNOWLEDGE CROSSWORDS DETAILS

In this section, we elaborate on the details of benchmark construction.

1. YAGO filtering
    (a) We remove edges in YAGO with the following relations: (i) Location-related: isLocatedIn, livesIn, happenedIn, diedIn, wasBornIn; (ii) Time-sensitive: worksAt, playsFor, isAffili-

| Hyperparameter | Value |
|---|---|
| $\text{degree}_c$ | 5, 7, 9 |
| $s_G$ | 6, 7, 8, 9, 10, 11 |
| $s_B$ | $[\frac{1}{4} \cdot s_G, \frac{1}{2} \cdot s_G]$ |
| $m_r$ | 1.1, 1.2, 1.3 |
| $m_b$ | 1, 1.1 |

Table 7: Hyperparamters for benchmark construction

    atedTo, isPoliticianOf, isLeaderOf; (iii) Not self-evident: influences, owns, isKnownFor, dealsWith, imports, exports, created, isInterestedIn, dealsWith, isConnectedTo.

    (b) The remaining relations in YAGO are: graduatedFrom, hasMusicalRole, hasAcademicAdvisor, hasChild, wroteMusicFor, hasCapital, actedIn, hasOfficialLanguage, hasWonPrize, hasGender, hasCurrency, directed, isCitizenOf, participatedIn, isMarriedTo, hasNeighbor, edited.

2. Modified k-hop neighborhood

    (a) A center node $c$ is randomly selected from nodes with degree $\text{degree}_c$ in filtered YAGO.

    (b) We retrieve a modified 5-hop neighborhood $\mathcal{G}_{\mathcal{N}}^{(c)}$: in each layer, we retain at most 8 nodes. This approach assists us in obtaining a subgraph with a relatively long diameter while avoiding excessive density.

3. Downsample to $\mathcal{G}_{\mathcal{A}}$

    (a) We repeatedly remove nodes from $\mathcal{G}_{\mathcal{N}}^{(c)}$ until the number of nodes in the largest connected component in $\mathcal{G}_{\mathcal{N}}^{(c)}$ is no more than question graph size $s_G$.

        i. Sort the nodes in $\mathcal{G}_{\mathcal{N}}^{(c)}$ by degree in filtered YAGO in descending order as $\mathbf{v}_{\text{sorted,YAGO}}$.

        ii. Denote *reduce range multiplier* as $m_r$. Then *reduce range rr* is calculated as $m_r \cdot (|\mathcal{V}_{\mathcal{N}^{(c)}}| - s_G)$ where $\mathcal{V}_{\mathcal{N}^{(c)}}$ is the set of nodes in $\mathcal{G}_{\mathcal{N}}^{(c)}$.

        iii. Randomly pick a node in $\mathbf{v}_{\text{sorted,YAGO}}[(rr-1)/2 : (rr-1)]$ and remove this node from $\mathcal{G}_{\mathcal{N}}^{(c)}$.

    (b) Following the abovementioned approach, we downsample $\mathcal{G}_{\mathcal{N}}^{(c)}$ to $\mathcal{G}_{\mathcal{A}}$ by removing nodes with relatively high degree in filtered YAGO and introduce randomness in this process.

4. Generate blanks to get $\mathcal{G}_{\mathcal{Q}}$

    (a) To mask $s_B$ nodes in $\mathcal{G}_{\mathcal{A}}$ as blanks, denote *blank range multiplier* as $m_b$ and calculate *blank range br* as $s_B \cdot m_b$.

    (b) Sort the nodes in $\mathcal{G}_{\mathcal{N}}^{(c)}$ by degree in $\mathcal{G}_{\mathcal{Q}}$ in descending order as $\mathbf{v}_{\text{sorted},\mathcal{G}_{\mathcal{A}}}$.

    (c) We then randomly select $s_B$ nodes from the first $br$ nodes in $\mathbf{v}_{\text{sorted},\mathcal{G}_{\mathcal{A}}}$ as blanks.

Specifically, the hyperparameters we used for benchmark construction are listed in Tabel 7.

# D  QUALITATIVE ANALYSIS

In this section, we provide examples of knowledge crosswords that GPT-3.5-TURBO answers correctly or wrongly using STAGED PROMPTING and VERIFY-ALL. In-context exemplars are omitted in this section to save space and can be found in Appendix E. Table 8 and Table 9 show results using STAGED PROMPTING; Table 10, Table 11 and Table 12 show results using VERIFY-ALL.

Table 8: Response using STAGED PROMPTING where the answers are correct.

| STAGED PROMPTING | |
|---|---|
| **Prompt** | **Response** |
| [STAGED PROMPTING exemplars] Instruction: Pick the correct answer for each blank that satisfies all the given constraints. Knowledge: (June Allyson, is married to, Dick Powell); (Bradley Cooper, is married to, Jennifer Esposito); (Charlie Chaplin, is married to, Oona O'Neill); (Joan Blondell, is married to, Dick Powell); (Joan Blondell, acted in, Support Your Local Gunfighter); (Suzanne Pleshette, acted in, Support Your Local Gunfighter); (James Garner, acted in, Support Your Local Gunfighter); (Jack Elam, acted in, Support Your Local Gunfighter); (Dick Powell, is married to, June Allyson); (Bradley Cooper, is married to, Jennifer Esposito); (Charlie Chaplin, is married to, Oona O'Neill); (Dick Powell, is married to, Joan Blondell); (Dick Powell, has gender, male); (Eric Maskin, has gender, male); (Charlie Chaplin, has gender, male); (Robert Bly, has gender, male); (Eric Maskin, has gender, male); (Robert Bly, has gender, male); (Charlie Chaplin, has gender, male); (Borislav Mikhailov, has gender, male); (Charlie Chaplin, has gender, male); (Eric Maskin, has gender, male); (Robert Bly, has gender, male); (Cole Tinkler, has gender, male). Constraints: (blank 2, is married to, Dick Powell); (blank 2, acted in, Support Your Local Gunfighter); (Dick Powell, is married to, blank 2); (Dick Powell, has gender, blank 1); (Borislav Mikhailov, has gender, blank 1); (Cole Tinkler, has gender, blank 1). Options: blank 1: A. female, B. male blank 2: A. Suzanne Pleshette, B. Joan Blondell, C. James Garner Answer: | Stage 1 - status: Current answer: None. Remaining constraints containing blanks: (blank 2, is married to, Dick Powell); (blank 2, acted in, Support Your Local Gunfighter); (Dick Powell, is married to, blank 2); (Dick Powell, has gender, blank 1); (Borislav Mikhailov, has gender, blank 1); (Cole Tinkler, has gender, blank 1). Stage 1 - solve: From (Dick Powell, has gender, blank 1), candidate for blank 1: B. male. Stage 1 - status update: Filled remaining constraints with current answer: (Dick Powell, has gender, male); (Borislav Mikhailov, has gender, male); (Cole Tinkler, has gender, male). Updated remaining constraints containing blanks: (blank 2, is married to, Dick Powell); (blank 2, acted in, Support Your Local Gunfighter); (Dick Powell, is married to, blank 2). Stage1 - verify filled constraints: Does any error occur in filled remaining constraints? No. Go to next stage. Stage 2 - status: Current answer: blank 1: B. male. Remaining constraints containing blanks: (blank 2, is married to, Dick Powell); (blank 2, acted in, Support Your Local Gunfighter); (Dick Powell, is married to, blank 2). Stage 2 - solve: From (blank 2, is married to, Dick Powell), candidate for blank 2: B. Joan Blondell. Stage 2 - status update: Filled remaining constraints with current answer: (Joan Blondell, is married to, Dick Powell); (Joan Blondell, acted in, Support Your Local Gunfighter); (Dick Powell, is married to, Joan Blondell). Updated remaining constraints containing blanks: Stage2 - verify filled constraints: Does any error occur in filled remaining constraints? No. Go to next stage. Stage 3 - status: Current answer: blank 1: B. male, blank 2: B. Joan Blondell. Remaining constraints with blanks: None. Stage 3 - solve: No more remaining constraints with blank. Final answer: blank 1: B, blank 2: B ✓ |

Table 9: Response using STAGED PROMPTING where the answers are wrong.

| STAGED PROMPTING | |
|---|---|
| **Prompt** | **Response** |
| [STAGED PROMPTING exemplars] Instruction: Pick the correct answer for each blank that satisfies all the given constraints. Knowledge: (Paz Vega, acted in, The Human Contract); (Joe Roberts, acted in, Our Hospitality); (Paz Vega, acted in, The Six Wives of Henry Lefay); (Paz Vega, acted in, The Spirit (film)); (Charlie Chaplin, directed, A Woman of Paris); (Charlie Chaplin, directed, The Champion (1915 film)); (Charlie Chaplin, directed, Limelight (1952 film)); (Jada Pinkett Smith, directed, The Human Contract); (Johnathon Schaech, acted in, Prom Night (2008 film)); (Joe Roberts, acted in, Our Hospitality); (Brittany Snow, acted in, Prom Night (2008 film)); (Idris Elba, acted in, Prom Night (2008 film)); (Joe Roberts, acted in, The Haunted House (1921 film)); (Idris Elba, acted in, The Human Contract); (Joe Roberts, acted in, Our Hospitality); (William Hopper, acted in, The Bad Seed (1956 film)). Constraints: (Paz Vega, acted in, blank 1); (Jada Pinkett Smith, directed, blank 1); (blank 2, acted in, Prom Night (2008 film)); (blank 2, acted in, blank 1). Options: blank 1: A. The Human Contract, B. The Spirit (film), C. The Six Wives of Henry Lefay blank 2: A. Johnathon Schaech, B. Idris Elba, C. Brittany Snow Answer: | Stage 1 - status: Current answer: None. Remaining constraints containing blanks: (Paz Vega, acted in, blank 1); (Jada Pinkett Smith, directed, blank 1); (blank 2, acted in, Prom Night (2008 film)); (blank 2, acted in, blank 1). Stage 1 - solve: From (Paz Vega, acted in, blank 1), candidate for blank 1: A. The Human Contract. Stage 1 - status update: Filled remaining constraints with current answer: (Paz Vega, acted in, The Human Contract); (Jada Pinkett Smith, directed, The Human Contract). Updated remaining constraints containing blanks: (blank 2, acted in, Prom Night (2008 film)); (blank 2, acted in, The Human Contract). Stage1 - verify filled constraints: Does any error occur in filled remaining constraints? No. Go to next stage. Stage 2 - status: Current answer: blank 1: A. The Human Contract. Remaining constraints containing blanks: (blank 2, acted in, Prom Night (2008 film)); (blank 2, acted in, The Human Contract). Stage 2 - solve: From (blank 2, acted in, Prom Night (2008 film)), candidate for blank 2: A. Johnathon Schaech. Stage 2 - status update: Filled remaining constraints with current answer: (Johnathon Schaech, acted in, Prom Night (2008 film)); (Johnathon Schaech, acted in, The Human Contract). Updated remaining constraints containing blanks: Stage2 - verify filled constraints: Does any error occur in filled remaining constraints? No. Go to next stage. Stage 3 - status: Current answer: blank 1: A. The Human Contract, blank 2: A. Johnathon Schaech. Remaining constraints with blanks: None. Stage 3 - solve: No more remaining constraints with blank. Final answer: blank 1: A, blank 2: A ✗ (Correct answer: blank 1: A, blank 2: B) |

Table 10: Response using VERIFY-ALL where the answers are correct and involves error detection and backtracking.

| VERIFY-ALL | |
|---|---|
| **Prompt** | **Response** |
| [VERIFY-ALL exemplars] Instruction: Pick the correct answer for each blank that satisfies all the given constraints. Knowledge: (June Allyson, is married to, Dick Powell); (Bradley Cooper, is married to, Jennifer Esposito); (Charlie Chaplin, is married to, Oona O'Neill); (Joan Blondell, is married to, Dick Powell); (Joan Blondell, acted in, Support Your Local Gunfighter); (Suzanne Pleshette, acted in, Support Your Local Gunfighter); (James Garner, acted in, Support Your Local Gunfighter); (Jack Elam, acted in, Support Your Local Gunfighter); (Dick Powell, is married to, June Allyson); (Bradley Cooper, is married to, Jennifer Esposito); (Charlie Chaplin, is married to, Oona O'Neill); (Dick Powell, is married to, Joan Blondell); (Dick Powell, has gender, male); (Eric Maskin, has gender, male); (Charlie Chaplin, has gender, male); (Robert Bly, has gender, male); (Eric Maskin, has gender, male); (Robert Bly, has gender, male); (Charlie Chaplin, has gender, male); (Borislav Mikhailov, has gender, male); (Charlie Chaplin, has gender, male); (Eric Maskin, has gender, male); (Robert Bly, has gender, male); (Cole Tinkler, has gender, male). Constraints: (blank 2, is married to, Dick Powell); (blank 2, acted in, Support Your Local Gunfighter); (Dick Powell, is married to, blank 2); (Dick Powell, has gender, blank 1); (Borislav Mikhailov, has gender, blank 1); (Cole Tinkler, has gender, blank 1). Options: blank 1: A. female, B. male blank 2: A. Suzanne Pleshette, B. Joan Blondell, C. James Garner Answer: | Candidate answer: blank 1: A. female, blank 2: B. Joan Blondell; Filled constraints with candidate answer: (Joan Blondell, is married to, Dick Powell); (Joan Blondell, acted in, Support Your Local Gunfighter); (Dick Powell, is married to, Joan Blondell); (Dick Powell, has gender, female); (Borislav Mikhailov, has gender, female); (Cole Tinkler, has gender, female); Verification: Does error occur in filled constraints with candidate answer? (Dick Powell, has gender, female) is incorrect. Candidate answer: blank 1: B. male, blank 2: B. Joan Blondell; Filled constraints with candidate answer: (Joan Blondell, is married to, Dick Powell); (Joan Blondell, acted in, Support Your Local Gunfighter); (Dick Powell, is married to, Joan Blondell); (Dick Powell, has gender, male); (Borislav Mikhailov, has gender, male); (Cole Tinkler, has gender, male); Verification: Does error occur in filled constraints with candidate answer? No. Therefore, blank 1: B, blank 2: B. ✓ |

Table 11: Response using VERIFY-ALL where the answers are correct correctly in a single trial.

| VERIFY-ALL | |
| --- | --- |
| **Prompt** | **Response** |
| [VERIFY-ALL exemplars] Instruction: Pick the correct answer for each blank that satisfies all the given constraints. Knowledge: (Paz Vega, acted in, The Human Contract); (Joe Roberts, acted in, Our Hospitality); (Paz Vega, acted in, The Six Wives of Henry Lefay); (Paz Vega, acted in, The Spirit (film)); (Charlie Chaplin, directed, A Woman of Paris); (Charlie Chaplin, directed, The Champion (1915 film)); (Charlie Chaplin, directed, Limelight (1952 film)); (Jada Pinkett Smith, directed, The Human Contract); (Johnathon Schaech, acted in, Prom Night (2008 film)); (Joe Roberts, acted in, Our Hospitality); (Brittany Snow, acted in, Prom Night (2008 film)); (Idris Elba, acted in, Prom Night (2008 film)); (Joe Roberts, acted in, The Haunted House (1921 film)); (Idris Elba, acted in, The Human Contract); (Joe Roberts, acted in, Our Hospitality); (William Hopper, acted in, The Bad Seed (1956 film)). Constraints: (Paz Vega, acted in, blank 1); (Jada Pinkett Smith, directed, blank 1); (blank 2, acted in, Prom Night (2008 film)); (blank 2, acted in, blank 1). Options: blank 1: A. The Human Contract, B. The Spirit (film), C. The Six Wives of Henry Lefay blank 2: A. Johnathon Schaech, B. Idris Elba, C. Brittany Snow Answer: | Candidate answer: blank 1: A. The Human Contract, blank 2: B. Idris Elba; Filled constraints with candidate answer: (Paz Vega, acted in, The Human Contract); (Jada Pinkett Smith, directed, The Human Contract); (Idris Elba, acted in, Prom Night (2008 film)); (Idris Elba, acted in, The Human Contract); Verification: Does error occur in filled constraints with candidate answer? No. Therefore, blank 1: A, blank 2: B. ✓ |

Table 12: Response using VERIFY-ALL where the answers are wrong.

| VERIFY-ALL | |
| --- | --- |
| **Prompt** | **Response** |
| [VERIFY-ALL exemplars] Instruction: Pick the correct answer for each blank that satisfies all the given constraints. Knowledge: (Colleen Camp, has gender, female); (Robert Bly, has gender, male); (Eric Maskin, has gender, male); (Charlie Chaplin, has gender, male); (Grace Zabriskie, has gender, female); (Robert Bly, has gender, male); (Eric Maskin, has gender, male); (Charlie Chaplin, has gender, male); (Tim Conway, acted in, The Private Eyes (1980 film)); (Don Knotts, acted in, The Private Eyes (1980 film)); (Bernard Fox (actor), acted in, The Private Eyes (1980 film)); (Grace Zabriskie, acted in, The Private Eyes (1980 film)); (Charlie Chaplin, has gender, male); (Eric Maskin, has gender, male); (Diane Baker, has gender, female); (Robert Bly, has gender, male); (Jane Seymour, has gender, female); (Eric Maskin, has gender, male); (Robert Bly, has gender, male); (Charlie Chaplin, has gender, male). Constraints: (Colleen Camp, has gender, blank 2); (blank 1, has gender, blank 2); (blank 1, acted in, The Private Eyes (1980 film)); (Diane Baker, has gender, blank 2); (Jane Seymour, has gender, blank 2). Options: blank 1: A. Bernard Fox (actor), B. Grace Zabriskie, C. Don Knotts blank 2: A. female, B. male Answer: | Candidate answer: blank 1: C. Don Knotts, blank 2: A. female; Filled constraints with candidate answer: (Colleen Camp, has gender, female); (Don Knotts, has gender, female); (Don Knotts, acted in, The Private Eyes (1980 film)); (Diane Baker, has gender, female); (Jane Seymour, has gender, female); Verification: Does error occur in filled constraints with candidate answer? No. Therefore, blank 1: C, blank 2: A. ✗ (Correct answer: blank 1: B, blank 2: A) |

# E  PROMPTS

We list the prompts for all experiments in Tables 2, 3 and 6.

Table 13: Prompts used in main experiments for all baseline approaches and our approaches, with exemplars and an example question. "Knowledge" for each problem is only applicable in the "w/ Relevant Knowledge" setting.

| An example of knowledge crossword |
|---|
| Instruction: Pick the correct answer for each blank that satisfies all the given constraints.
Desired format: blank i: Z ...
Constraints: (Paz Vega, acted in, blank 1); (Jada Pinkett Smith, directed, blank 1); (blank 2, acted in, Prom Night (2008 film)); (blank 2, acted in, blank 1).
Options:
blank 1: A. The Human Contract, B. The Spirit (film), C. The Six Wives of Henry Lefay
blank 2: A. Johnathon Schaech, B. Idris Elba, C. Brittany Snow
Answer: blank 1: A, blank 2: B |

| Method | Prompt |
|---|---|
| UPPERBOUND | Instruction: Pick the correct answer for each blank that satisfies all the given constraints.
Desired format: blank i: Z ...
Knowledge: (Paz Vega, acted in, The Human Contract); (Jada Pinkett Smith, directed, The Human Contract); (Idris Elba, acted in, Prom Night (2008 film)); (Idris Elba, acted in, The Human Contract). Constraints: (Paz Vega, acted in, blank 1); (Jada Pinkett Smith, directed, blank 1); (blank 2, acted in, Prom Night (2008 film)); (blank 2, acted in, blank 1).
Options: blank 1: A. The Human Contract, B. The Spirit (film), C. The Six Wives of Henry Lefay blank 2: A. Johnathon Schaech, B. Idris Elba, C. Brittany Snow
Answer: |
| ZERO-SHOT | Instruction: Pick the correct answer for each blank that satisfies all the given constraints.
Knowledge: (Paz Vega, acted in, The Human Contract); (Joe Roberts, acted in, Our Hospitality); (Paz Vega, acted in, The Six Wives of Henry Lefay); (Paz Vega, acted in, The Spirit (film)); (Charlie Chaplin, directed, A Woman of Paris); (Charlie Chaplin, directed, The Champion (1915 film)); (Charlie Chaplin, directed, Limelight (1952 film)); (Jada Pinkett Smith, directed, The Human Contract); (Johnathon Schaech, acted in, Prom Night (2008 film)); (Joe Roberts, acted in, Our Hospitality); (Brittany Snow, acted in, Prom Night (2008 film)); (Idris Elba, acted in, Prom Night (2008 film)); (Joe Roberts, acted in, The Haunted House (1921 film)); (Idris Elba, acted in, The Human Contract); (Joe Roberts, acted in, Our Hospitality); (William Hopper, acted in, The Bad Seed (1956 film)). (Optional)
Desired format: blank i: Z Constraints: (Paz Vega, acted in, blank 1); (Jada Pinkett Smith, directed, blank 1); (blank 2, acted in, Prom Night (2008 film)); (blank 2, acted in, blank 1).
Options: blank 1: A. The Human Contract, B. The Spirit (film), C. The Six Wives of Henry Lefay blank 2: A. Johnathon Schaech, B. Idris Elba, C. Brittany Snow
Answer: |

FEW-SHOT

Instruction: Pick the correct answer for each blank that satisfies all the given constraints.
Knowledge: (Charlton Heston, acted in, True Lies); (Eliza Dushku, acted in, True Lies); (Tom Arnold (actor), acted in, True Lies); (Bill Paxton, acted in, True Lies); (Charlton Heston, acted in, Chiefs (miniseries)); (Stephen Collins, acted in, Chiefs (miniseries)); (Paul Sorvino, acted in, Chiefs (miniseries)); (Danny Glover, acted in, Chiefs (miniseries)).
(Optional) Constraints: (blank 1, acted in, True Lies); (blank 1, acted in, Chiefs (miniseries)).
Options: blank 1: A. Bill Paxton, B. Charlton Heston, C. Paul Sorvino
Answer: blank 1: B
Instruction: Pick the correct answer for each blank that satisfies all the given constraints.
Knowledge: (Joe Roberts, acted in, Our Hospitality); (Joe Roberts, acted in, Neighbors (1920 film)); (Taye Diggs, acted in, Rent (film)); (Joe Roberts, acted in, Three Ages); (Bradley Cooper, is married to, Jennifer Esposito); (Taye Diggs, is married to, Idina Menzel); (Charlie Chaplin, is married to, Mildred Harris); (Mary, Queen of Hungary, is married to, Jobst of Moravia); (Idina Menzel, acted in, Enchanted (film)); (Idina Menzel, acted in, Rent (film)); (Joe Roberts, acted in, Neighbors (1920 film)); (Joe Roberts, acted in, Three Ages); (Idina Menzel, is married to, Taye Diggs); (Bradley Cooper, is married to, Jennifer Esposito); (Mary, Queen of Hungary, is married to, Jobst of Moravia); (Charlie Chaplin, is married to, Mildred Harris). (Optional)
Constraints: (blank 1, acted in, blank 2); (blank 1, is married to, Idina Menzel); (Idina Menzel, acted in, blank 2); (Idina Menzel, is married to, blank 1).
Options: blank 1: A. Kelly LeBrock, B. Napoleon, C. Taye Diggs blank 2: A. Halloweentown High, B. Magnolia (film), C. Rent (film)
Answer: blank 1: C, blank 2: C
Instruction: Pick the correct answer for each blank that satisfies all the given constraints.
Knowledge: (Andy García, acted in, Smokin' Aces); (Andy García, acted in, Ocean's Thirteen); (Andy García, acted in, The Untouchables (film)); (Andy García, acted in, The Pink Panther 2); (Jeremy Piven, acted in, Smokin' Aces); (Joe Roberts, acted in, Our Hospitality); (Joe Roberts, acted in, Three Ages); (Joe Roberts, acted in, Neighbors (1920 film)); (Virginia Madsen, acted in, Scooby-Doo! in Where's My Mummy?); (Jeremy Piven, acted in, Scooby-Doo! in Where's My Mummy?); (Mindy Cohn, acted in, Scooby-Doo! in Where's My Mummy?); (Grey DeLisle, acted in, Scooby-Doo! in Where's My Mummy?). (Optional)
Constraints: (Andy García, acted in, blank 1); (blank 2, acted in, blank 1); (blank 2, acted in, Scooby-Doo! in Where's My Mummy?).
Options: blank 1: A. Things to Do in Denver When You're Dead, B. Smokin' Aces, C. Beverly Hills Chihuahua blank 2: A. Ron Perlman, B. Casey Kasem, C. Jeremy Piven
Answer: blank 1: B, blank 2: C
Instruction: Pick the correct answer for each blank that satisfies all the given constraints.
Knowledge: (Miriam Margolyes, acted in, Harry Potter (film series)); (David Thewlis, acted in, Harry Potter (film series)); (John Cleese, acted in, Harry Potter (film series)); (Richard Harris, acted in, Harry Potter (film series)); (Joe Roberts, acted in, Three Ages); (Maureen Lipman, acted in, A Little Princess (1986 TV serial)); (Nigel Havers, acted in, A Little Princess (1986 TV serial)); (Miriam Margolyes, acted in, A Little Princess (1986 TV serial)). (Optional)
Constraints: (blank 1, acted in, Harry Potter (film series)); (blank 1, acted in, A Little Princess (1986 TV serial)).
Options: blank 1: A. Miriam Margolyes, B. Maggie Smith, C. Emma Watson
Answer: blank 1: A
Instruction: Pick the correct answer for each blank that satisfies all the given constraints. Knowledge: (Joe Roberts, acted in, Neighbors (1920 film)); (Dinah Sheridan, acted in, The Railway Children (film)); (Joe Roberts, acted in, Three Ages); (Dinah Sheridan, acted in, 29 Acacia Avenue); (Dinah Sheridan, is married to, John Merivale); (Charlie Chaplin, is married to, Mildred Harris); (Dinah Sheridan, is married to, Jimmy Hanley); (Bradley Cooper, is married to, Jennifer Esposito); (Joe Roberts, acted in, Neighbors (1920 film)); (Jimmy Hanley, acted in, 29 Acacia Avenue); (Joe Roberts, acted in, Three Ages); (Joe Roberts, acted in, Our Hospitality); (Charlie Chaplin, is married to, Mildred Harris); (Dinah Sheridan, is married to, Jimmy Hanley); (Charlie Chaplin, is married to, Oona O'Neill); (Dinah Sheridan, is married to, John Merivale); (Charlie Chaplin, is married to, Mildred Harris); (Charlie Chaplin, is married to, Oona O'Neill); (John Merivale, is married to, Jan Sterling); (Paul Douglas (actor), is married to, Jan Sterling).
Constraints: (Dinah Sheridan, acted in, blank 2); (Dinah Sheridan, is married to, blank 1); (blank 1, acted in, blank 2); (Dinah Sheridan, is married to, blank 3); (blank 3, is married to, Jan Sterling). (Optional)
Constraints: (Dinah Sheridan, acted in, blank 2); (Dinah Sheridan, is married to, blank 1); (blank 1, acted in, blank 2); (Dinah Sheridan, is married to, blank 3); (blank 3, is married to, Jan Sterling).
Options: blank 1: A. Liam Neeson, B. Jimmy Hanley, C. Nancy Wilson (rock musician) blank 2: A. Courage of Lassie, B. 29 Acacia Avenue, C. Listen to Me (film) blank 3: A. José María Aznar, B. John Merivale, C. Prince Harald of Denmark
Answer: blank 1: B, blank 2: B, blank 3: B
Instruction: Pick the correct answer for each blank that satisfies all the given constraints.
Knowledge: (Paz Vega, acted in, The Human Contract); (Joe Roberts, acted in, Our Hospitality); (Paz Vega, acted in, The Six Wives of Henry Lefay); (Paz Vega, acted in, The Spirit (film)); (Charlie Chaplin, directed, A Woman of Paris); (Charlie Chaplin, directed, The Champion (1915 film)); (Charlie Chaplin, directed, Limelight (1952 film)); (Jada Pinkett Smith, directed, The Human Contract); (Johnathon Schaech, acted in, Prom Night (2008 film)); (Joe Roberts, acted in, Our Hospitality); (Brittany Snow, acted in, Prom Night (2008 film)); (Idris Elba, acted in, Prom Night (2008 film)); (Joe Roberts, acted in, The Haunted House (1921 film)); (Idris Elba, acted in, The Human Contract); (Joe Roberts, acted in, Our Hospitality); (William Hopper, acted in, The Bad Seed (1956 film)). (Optional)
Constraints: (Paz Vega, acted in, blank 1); (Jada Pinkett Smith, directed, blank 1); (blank 2, acted in, Prom Night (2008 film)); (blank 2, acted in, blank 1).
Options: blank 1: A. The Human Contract, B. The Spirit (film), C. The Six Wives of Henry Lefay blank 2: A. Johnathon Schaech, B. Idris Elba, C. Brittany Snow
Answer:

CoT

Instruction: Pick the correct answer for each blank that satisfies all the given constraints.

Knowledge: (Charlton Heston, acted in, True Lies); (Eliza Dushku, acted in, True Lies); (Tom Arnold (actor), acted in, True Lies); (Bill Paxton, acted in, True Lies); (Charlton Heston, acted in, Chiefs (miniseries)); (Stephen Collins, acted in, Chiefs (miniseries)); (Paul Sorvino, acted in, Chiefs (miniseries)); (Danny Glover, acted in, Chiefs (miniseries)). (Optional)

Constraints: (blank 1, acted in, True Lies); (blank 1, acted in, Chiefs (miniseries)).

Options: blank 1: A. Bill Paxton, B. Charlton Heston, C. Paul Sorvino

Answer: (Charlton Heston, acted in, True Lies); (Charlton Heston, acted in, Chiefs (miniseries)). Therefore, blank 1: B

Instruction: Pick the correct answer for each blank that satisfies all the given constraints.

Knowledge: (Joe Roberts, acted in, Our Hospitality); (Joe Roberts, acted in, Neighbors (1920 film)); (Taye Diggs, acted in, Rent (film)); (Joe Roberts, acted in, Three Ages); (Bradley Cooper, is married to, Jennifer Esposito); (Taye Diggs, is married to, Idina Menzel); (Charlie Chaplin, is married to, Mildred Harris); (Mary, Queen of Hungary, is married to, Jobst of Moravia); (Idina Menzel, acted in, Enchanted (film)); (Idina Menzel, acted in, Rent (film)); (Joe Roberts, acted in, Neighbors (1920 film)); (Joe Roberts, acted in, Three Ages); (Idina Menzel, is married to, Taye Diggs); (Bradley Cooper, is married to, Jennifer Esposito); (Mary, Queen of Hungary, is married to, Jobst of Moravia); (Charlie Chaplin, is married to, Mildred Harris). (Optional)

Constraints: (blank 1, acted in, blank 2); (blank 1, is married to, Idina Menzel); (Idina Menzel, acted in, blank 2); (Idina Menzel, is married to, blank 1).

Options: blank 1: A. Kelly LeBrock, B. Napoleon, C. Taye Diggs blank 2: A. Halloweentown High, B. Magnolia (film), C. Rent (film)

Answer: (Taye Diggs, acted in, Rent (film)); (Taye Diggs, is married to, Idina Menzel); (Idina Menzel, acted in, Rent (film)). Therefore, blank 1: C, blank 2: C

Instruction: Pick the correct answer for each blank that satisfies all the given constraints.

Knowledge: (Andy García, acted in, Smokin' Aces); (Andy García, acted in, Ocean's Thirteen); (Andy García, acted in, The Untouchables (film)); (Andy García, acted in, The Pink Panther 2); (Jeremy Piven, acted in, Smokin' Aces); (Joe Roberts, acted in, Our Hospitality); (Joe Roberts, acted in, Three Ages); (Joe Roberts, acted in, Neighbors (1920 film)); (Virginia Madsen, acted in, Scooby-Doo! in Where's My Mummy?); (Jeremy Piven, acted in, Scooby-Doo! in Where's My Mummy?); (Mindy Cohn, acted in, Scooby-Doo! in Where's My Mummy?); (Grey DeLisle, acted in, Scooby-Doo! in Where's My Mummy?). (Optional)

Constraints: (Andy García, acted in, blank 1); (blank 2, acted in, blank 1); (blank 2, acted in, Scooby-Doo! in Where's My Mummy?).

Options: blank 1: A. Things to Do in Denver When You're Dead, B. Smokin' Aces, C. Beverly Hills Chihuahua blank 2: A. Ron Perlman, B. Casey Kasem, C. Jeremy Piven

Answer: (Andy García, acted in, Smokin' Aces); (Jeremy Piven, acted in, Smokin' Aces); (Jeremy Piven, acted in, Scooby-Doo! in Where's My Mummy?). Therefore, blank 1: B, blank 2: C

Instruction: Pick the correct answer for each blank that satisfies all the given constraints.

Knowledge: (Miriam Margolyes, acted in, Harry Potter (film series)); (David Thewlis, acted in, Harry Potter (film series)); (John Cleese, acted in, Harry Potter (film series)); (Richard Harris, acted in, Harry Potter (film series)); (Joe Roberts, acted in, Three Ages); (Maureen Lipman, acted in, A Little Princess (1986 TV serial)); (Nigel Havers, acted in, A Little Princess (1986 TV serial)); (Miriam Margolyes, acted in, A Little Princess (1986 TV serial)). (Optional)

Constraints: (blank 1, acted in, Harry Potter (film series)); (blank 1, acted in, A Little Princess (1986 TV serial)).

Options: blank 1: A. Miriam Margolyes, B. Maggie Smith, C. Emma Watson

Answer: (Miriam Margolyes, acted in, Harry Potter (film series)); (Miriam Margolyes, acted in, A Little Princess (1986 TV serial)). Therefore, blank 1: A

Instruction: Pick the correct answer for each blank that satisfies all the given constraints.

Knowledge: (Joe Roberts, acted in, Neighbors (1920 film)); (Dinah Sheridan, acted in, The Railway Children (film)); (Joe Roberts, acted in, Three Ages); (Dinah Sheridan, acted in, 29 Acacia Avenue); (Dinah Sheridan, is married to, John Merivale); (Charlie Chaplin, is married to, Mildred Harris); (Dinah Sheridan, is married to, Jimmy Hanley); (Bradley Cooper, is married to, Jennifer Esposito); (Joe Roberts, acted in, Neighbors (1920 film)); (Jimmy Hanley, acted in, 29 Acacia Avenue); (Joe Roberts, acted in, Three Ages); (Joe Roberts, acted in, Our Hospitality); (Charlie Chaplin, is married to, Mildred Harris); (Dinah Sheridan, is married to, Jimmy Hanley); (Charlie Chaplin, is married to, Oona O'Neill); (Dinah Sheridan, is married to, John Merivale); (Charlie Chaplin, is married to, Mildred Harris); (Charlie Chaplin, is married to, Oona O'Neill); (John Merivale, is married to, Jan Sterling); (Paul Douglas (actor), is married to, Jan Sterling).

Constraints: (Dinah Sheridan, acted in, blank 2); (Dinah Sheridan, is married to, blank 1); (blank 1, acted in, blank 2); (Dinah Sheridan, is married to, blank 3); (blank 3, is married to, Jan Sterling). (Optional)

Constraints: (Dinah Sheridan, acted in, blank 2); (Dinah Sheridan, is married to, blank 1); (blank 1, acted in, blank 2); (Dinah Sheridan, is married to, blank 3); (blank 3, is married to, Jan Sterling).

Options: blank 1: A. Liam Neeson, B. Jimmy Hanley, C. Nancy Wilson (rock musician) blank 2: A. Courage of Lassie, B. 29 Acacia Avenue, C. Listen to Me (film) blank 3: A. José María Aznar, B. John Merivale, C. Prince Harald of Denmark

Answer: (Dinah Sheridan, acted in, 29 Acacia Avenue); (Dinah Sheridan, is married to, Jimmy Hanley); (Jimmy Hanley, acted in, 29 Acacia Avenue); (Dinah Sheridan, is married to, John Merivale); (John Merivale, is married to, Jan Sterling). Therefore, blank 1: B, blank 2: B, blank 3: B

Instruction: Pick the correct answer for each blank that satisfies all the given constraints.

Knowledge: (Paz Vega, acted in, The Human Contract); (Joe Roberts, acted in, Our Hospitality); (Paz Vega, acted in, The Six Wives of Henry Lefay); (Paz Vega, acted in, The Spirit (film)); (Charlie Chaplin, directed, A Woman of Paris); (Charlie Chaplin, directed, The Champion (1915 film)); (Charlie Chaplin, directed, Limelight (1952 film)); (Jada Pinkett Smith, directed, The Human Contract); (Johnathon Schaech, acted in, Prom Night (2008 film)); (Joe Roberts, acted in, Our Hospitality); (Brittany Snow, acted in, Prom Night (2008 film)); (Joe Roberts, acted in, The Haunted House (1921 film)); (Idris Elba, acted in, The Human Contract); (Joe Roberts, acted in, Our Hospitality); (William Hopper, acted in, The Bad Seed (1956 film)). (Optional)

Constraints: (Paz Vega, acted in, blank 1); (Jada Pinkett Smith, directed, blank 1); (blank 2, acted in, Prom Night (2008 film)); (blank 2, acted in, blank 1).

Options: blank 1: A. The Human Contract, B. The Spirit (film), C. The Six Wives of Henry Lefay blank 2: A. Johnathon Schaech, B. Idris Elba, C. Brittany Snow

Answer:

LTM

Instruction: Pick the correct answer for each blank that satisfies all the given constraints.

Knowledge: (Charlton Heston, acted in, True Lies); (Eliza Dushku, acted in, True Lies); (Tom Arnold (actor), acted in, True Lies); (Bill Paxton, acted in, True Lies); (Charlton Heston, acted in, Chiefs (miniseries)); (Stephen Collins, acted in, Chiefs (miniseries)); (Paul Sorvino, acted in, Chiefs (miniseries)); (Danny Glover, acted in, Chiefs (miniseries)). (Optional)

Constraints: (blank 1, acted in, True Lies); (blank 1, acted in, Chiefs (miniseries)).

Options: blank 1: A. Bill Paxton, B. Charlton Heston, C. Paul Sorvino

Answer: Considering (blank 1, acted in, True Lies), maybe blank 1: A, or blank 1: B; considering (blank 1, acted in, True Lies), (blank 1, acted in, Chiefs (miniseries)), maybe blank 1: B. Therefore, blank 1: B

Instruction: Pick the correct answer for each blank that satisfies all the given constraints.

Knowledge: (Joe Roberts, acted in, Our Hospitality); (Joe Roberts, acted in, Neighbors (1920 film)); (Taye Diggs, acted in, Rent (film)); (Joe Roberts, acted in, Three Ages); (Bradley Cooper, is married to, Jennifer Esposito); (Taye Diggs, is married to, Idina Menzel); (Charlie Chaplin, is married to, Mildred Harris); (Mary, Queen of Hungary, is married to, Jobst of Moravia); (Idina Menzel, acted in, Enchanted (film)); (Idina Menzel, acted in, Rent (film)); (Joe Roberts, acted in, Neighbors (1920 film)); (Joe Roberts, acted in, Three Ages); (Idina Menzel, is married to, Taye Diggs); (Bradley Cooper, is married to, Jennifer Esposito); (Mary, Queen of Hungary, is married to, Jobst of Moravia); (Charlie Chaplin, is married to, Mildred Harris). (Optional)

Constraints: (blank 1, acted in, blank 2); (blank 1, is married to, Idina Menzel); (Idina Menzel, acted in, blank 2); (Idina Menzel, is married to, blank 1).

Options: blank 1: A. Kelly LeBrock, B. Napoleon, C. Taye Diggs blank 2: A. Halloweentown High, B. Magnolia (film), C. Rent (film)

Answer: Considering (blank 1, acted in, blank 2), maybe blank 1: C and blank 2: C; considering (blank 1, acted in, blank 2), (blank 1, is married to, Idina Menzel), maybe blank 1: C and blank 2: C; considering (blank 1, acted in, blank 2), (blank 1, is married to, Idina Menzel), (Idina Menzel, acted in, blank 2), maybe blank 1: C and blank 2: C; considering (blank 1, acted in, blank 2), (blank 1, is married to, Idina Menzel), (Idina Menzel, acted in, blank 2), (Idina Menzel, is married to, blank 1), maybe blank 1: C and blank 2: C; Therefore, blank 1: C, blank 2: C

Instruction: Pick the correct answer for each blank that satisfies all the given constraints.

Knowledge: (Andy García, acted in, Smokin' Aces); (Andy García, acted in, Ocean's Thirteen); (Andy García, acted in, The Untouchables (film)); (Andy García, acted in, The Pink Panther 2); (Jeremy Piven, acted in, Smokin' Aces); (Joe Roberts, acted in, Our Hospitality); (Joe Roberts, acted in, Three Ages); (Joe Roberts, acted in, Neighbors (1920 film)); (Virginia Madsen, acted in, Scooby-Doo! in Where's My Mummy?); (Jeremy Piven, acted in, Scooby-Doo! in Where's My Mummy?); (Mindy Cohn, acted in, Scooby-Doo! in Where's My Mummy?); (Grey DeLisle, acted in, Scooby-Doo! in Where's My Mummy?). (Optional)

Constraints: (Andy García, acted in, blank 1); (blank 2, acted in, blank 1); (blank 2, acted in, Scooby-Doo! in Where's My Mummy?).

Options: blank 1: A. Things to Do in Denver When You're Dead, B. Smokin' Aces, C. Beverly Hills Chihuahua blank 2: A. Ron Perlman, B. Casey Kasem, C. Jeremy Piven Answer: Considering (Andy García, acted in, blank 1), maybe blank 1: A, or blank 1: B, or blank 1: C; considering (Andy García, acted in, blank 1), (blank 2, acted in, blank 1), maybe blank 1: B and blank 2: C; considering (Andy García, acted in, blank 1), (blank 2, acted in, blank 1), (blank 2, acted in, Scooby-Doo! in Where's My Mummy?), maybe blank 1: B and blank 2: C. Therefore, blank 1: B, blank 2: C

Instruction: Pick the correct answer for each blank that satisfies all the given constraints.

Knowledge: (Miriam Margolyes, acted in, Harry Potter (film series)); (David Thewlis, acted in, Harry Potter (film series)); (John Cleese, acted in, Harry Potter (film series)); (Richard Harris, acted in, Harry Potter (film series)); (Joe Roberts, acted in, Three Ages); (Maureen Lipman, acted in, A Little Princess (1986 TV serial)); (Nigel Havers, acted in, A Little Princess (1986 TV serial)); (Miriam Margolyes, acted in, A Little Princess (1986 TV serial)). (Optional)

Constraints: (blank 1, acted in, Harry Potter (film series)); (blank 1, acted in, A Little Princess (1986 TV serial)).

Options: blank 1: A. Miriam Margolyes, B. Maggie Smith, C. Emma Watson

Answer: Considering (blank 1, acted in, Harry Potter (film series)), maybe blank 1: A, or blank 1: B, or blank 1: C; considering (blank 1, acted in, Harry Potter (film series)), (blank 1, acted in, A Little Princess (1986 TV serial)), maybe blank 1: A. Therefore, blank 1: A

Instruction: Pick the correct answer for each blank that satisfies all the given constraints.

Knowledge: (Joe Roberts, acted in, Neighbors (1920 film)); (Dinah Sheridan, acted in, The Railway Children (film)); (Joe Roberts, acted in, Three Ages); (Dinah Sheridan, acted in, 29 Acacia Avenue); (Dinah Sheridan, is married to, John Merivale); (Charlie Chaplin, is married to, Mildred Harris); (Dinah Sheridan, is married to, Jimmy Hanley); (Bradley Cooper, is married to, Jennifer Esposito); (Joe Roberts, acted in, Neighbors (1920 film)); (Jimmy Hanley, acted in, 29 Acacia Avenue); (Joe Roberts, acted in, Three Ages); (Joe Roberts, acted in, Our Hospitality); (Charlie Chaplin, is married to, Mildred Harris); (Dinah Sheridan, is married to, Jimmy Hanley); (Charlie Chaplin, is married to, Oona O'Neill); (Dinah Sheridan, is married to, John Merivale); (Charlie Chaplin, is married to, Mildred Harris); (Charlie Chaplin, is married to, Oona O'Neill); (John Merivale, is married to, Jan Sterling); (Paul Douglas (actor), is married to, Jan Sterling).

Constraints: (Dinah Sheridan, acted in, blank 2); (Dinah Sheridan, is married to, blank 1); (blank 1, acted in, blank 2); (Dinah Sheridan, is married to, blank 3); (blank 3, is married to, Jan Sterling). (Optional)

Constraints: (Dinah Sheridan, acted in, blank 2); (Dinah Sheridan, is married to, blank 1); (blank 1, acted in, blank 2); (Dinah Sheridan, is married to, blank 3); (blank 3, is married to, Jan Sterling).

Options: blank 1: A. Liam Neeson, B. Jimmy Hanley, C. Nancy Wilson (rock musician) blank 2: A. Courage of Lassie, B. 29 Acacia Avenue, C. Listen to Me (film) blank 3: A. José María Aznar, B. John Merivale, C. Prince Harald of Denmark

Answer: Considering (Dinah Sheridan, acted in, blank 2), maybe blank 2: B; considering (Dinah Sheridan, acted in, blank 2), (Dinah Sheridan, is married to, blank 1), maybe blank 1: B and blank 2: B; considering (Dinah Sheridan, acted in, blank 2), (Dinah Sheridan, is married to, blank 1), (blank 1, acted in, blank 2), maybe blank 1: B and blank 2: B; considering (Dinah Sheridan, acted in, blank 2), (Dinah Sheridan, is married to, blank 1), (blank 1, acted in, blank 2), (Dinah Sheridan, is married to, blank 3), maybe blank 1: B and blank 2: B and blank 3: B; considering (Dinah Sheridan, acted in, blank 2), (Dinah Sheridan, is married to, blank 1), (blank 1, acted in, blank 2), (Dinah Sheridan, is married to, blank 3), (blank 3, is married to, Jan Sterling), maybe blank 1: B and blank 2: B and blank 3: B. Therefore, blank 1: B, blank 2: B, blank 3: B

Instruction: Pick the correct answer for each blank that satisfies all the given constraints.

Knowledge: (Paz Vega, acted in, The Human Contract); (Joe Roberts, acted in, Our Hospitality); (Paz Vega, acted in, The Six Wives of Henry Lefay); (Paz Vega, acted in, The Spirit (film)); (Charlie Chaplin, directed, A Woman of Paris); (Charlie Chaplin, directed, The Champion (1915 film)); (Charlie Chaplin, directed, Limelight (1952 film)); (Jada Pinkett Smith, directed, The Human Contract); (Johnathon Schaech, acted in, Prom Night (2008 film)); (Joe Roberts, acted in, Our Hospitality); (Brittany Snow, acted in, Prom Night (2008 film)); (Idris Elba, acted in, Prom Night (2008 film)); (Joe Roberts, acted in, The Haunted House (1921 film)); (Idris Elba, acted in, The Human Contract); (Joe Roberts, acted in, Our Hospitality); (William Hopper, acted in, The Bad Seed (1956 film)). (Optional)

Constraints: (Paz Vega, acted in, blank 1); (Jada Pinkett Smith, directed, blank 1); (blank 2, acted in, Prom Night (2008 film)); (blank 2, acted in, blank 1).

Options: blank 1: A. The Human Contract, B. The Spirit (film), C. The Six Wives of Henry Lefay blank 2: A. Johnathon Schaech, B. Idris Elba, C. Brittany Snow

Answer:

STAGED PROMPTING (ours)

Instruction: Pick the correct answer for each blank that satisfies all the given constraints.

Knowledge: (Charlton Heston, acted in, True Lies); (Eliza Dushku, acted in, True Lies); (Tom Arnold (actor), acted in, True Lies); (Bill Paxton, acted in, True Lies); (Charlton Heston, acted in, Chiefs (miniseries)); (Stephen Collins, acted in, Chiefs (miniseries)); (Paul Sorvino, acted in, Chiefs (miniseries)); (Danny Glover, acted in, Chiefs (miniseries)). (Optional)

Constraints: (blank 1, acted in, Harry Potter (film series)); (blank 1, acted in, A Little Princess (1986 TV serial)).

Options: blank 1: A. Miriam Margolyes, B. Maggie Smith, C. Emma Watson

Answer: Stage 1 - status: Current answer: None. Remaining constraints containing blanks: (blank 1, acted in, Harry Potter (film series)); (blank 1, acted in, A Little Princess (1986 TV serial)). Stage 1 - solve: From (blank 1, acted in, Harry Potter (film series)), candidate for blank 1: A. Miriam Margolyes. Stage 1 - status update: Current answer: blank 1: A. Miriam Margolyes. Filled remaining constraints with current answer: (Miriam Margolyes, acted in, Harry Potter (film series)); (Miriam Margolyes, acted in, A Little Princess (1986 TV serial)). Updated remaining constraints containing blanks: None. Stage 1 - verify filled constraints: Does any error occur in filled remaining constraints? No. Go to next stage. Stage 2 - status: Current answer: blank 1: A. Miriam Margolyes. Remaining constraints containing blanks: None. Stage 2 - solve: No more remaining constraints with blank. Final answer: blank 1: A

Instruction: Pick the correct answer for each blank that satisfies all the given constraints.

Knowledge: (Miriam Margolyes, acted in, Harry Potter (film series)); (David Thewlis, acted in, Harry Potter (film series)); (John Cleese, acted in, Harry Potter (film series)); (Richard Harris, acted in, Harry Potter (film series)); (Joe Roberts, acted in, Three Ages); (Maureen Lipman, acted in, A Little Princess (1986 TV serial)); (Nigel Havers, acted in, A Little Princess (1986 TV serial)); (Miriam Margolyes, acted in, A Little Princess (1986 TV serial)). (Optional)

Constraints: (blank 1, acted in, True Lies); (blank 1, acted in, Chiefs (miniseries)).

Options: blank 1: A. Bill Paxton, B. Charlton Heston, C. Paul Sorvino

Answer: Stage 1 - status: Current answer: None. Remaining constraints containing blanks: (blank 1, acted in, True Lies); (blank 1, acted in, Chiefs (miniseries)). Stage 1 - solve: From (blank 1, acted in, True Lies), candidate for blank 1: A. Bill Paxton. Stage 1 - status update: Current answer: blank 1: A. Bill Paxton. Filled remaining constraints with current answer: (Bill Paxton, acted in, True Lies); (Bill Paxton, acted in, Chiefs (miniseries)). Updated remaining constraints containing blanks: None. Stage 1 - verify filled constraints: Does any error occur in filled remaining constraints? (Bill Paxton, acted in, Chiefs (miniseries)) is incorrect. Redo stage 1. Stage 1 - status: Current answer: None. Remaining constraints containing blanks: (blank 1, acted in, True Lies); (blank 1, acted in, Chiefs (miniseries)). Stage 1 - solve: From (blank 1, acted in, True Lies), candidate for blank 1: B. Charlton Heston. Stage 1 - status update: Current answer: blank 1: B. Charlton Heston. Filled remaining constraints with current answer: (Charlton Heston, acted in, True Lies); (Charlton Heston, acted in, Chiefs (miniseries)). Updated remaining constraints containing blanks: None. Stage 1 - verify filled constraints: Does any error occur in filled remaining constraints? No. Go to next stage. Stage 2 - status: Current answer: blank 1: B. Charlton Heston. Remaining constraints with blanks: None. Stage 2 - solve: No more remaining constraints with blank. Final answer: blank 1: B

Instruction: Pick the correct answer for each blank that satisfies all the given constraints.

Knowledge: (Andy García, acted in, Smokin' Aces); (Andy García, acted in, Ocean's Thirteen); (Andy García, acted in, The Untouchables (film)); (Andy García, acted in, The Pink Panther 2); (Jeremy Piven, acted in, Smokin' Aces); (Joe Roberts, acted in, Our Hospitality); (Joe Roberts, acted in, Three Ages); (Joe Roberts, acted in, Neighbors (1920 film)); (Virginia Madsen, acted in, Scooby-Doo! in Where's My Mummy?); (Jeremy Piven, acted in, Scooby-Doo! in Where's My Mummy?); (Mindy Cohn, acted in, Scooby-Doo! in Where's My Mummy?); (Grey DeLisle, acted in, Scooby-Doo! in Where's My Mummy?). (Optional)

Constraints: (Andy García, acted in, blank 1); (blank 2, acted in, blank 1); (blank 2, acted in, Scooby-Doo! in Where's My Mummy?).

Options: blank 1: A. Things to Do in Denver When You're Dead, B. Smokin' Aces, C. Beverly Hills Chihuahua blank 2: A. Ron Perlman, B. Casey Kasem, C. Jeremy Piven

Answer: Stage 1 - status: Current answer: None. Remaining constraints containing blanks: (Andy García, acted in, blank 1); (blank 2, acted in, blank 1); (blank 2, acted in, Scooby-Doo! in Where's My Mummy?). Stage 1 - solve: From (Andy García, acted in, blank 1), candidate for blank 1: A. Things to Do in Denver When You're Dead. Stage 1 - status update: Current answer: blank 1: A. Things to Do in Denver When You're Dead. Filled remaining constraints with current answer: (Andy García, acted in, Things to Do in Denver When You're Dead). Updated remaining constraints containing blanks: (blank 2, acted in, Things to Do in Denver When You're Dead); (blank 2, acted in, Scooby-Doo! in Where's My Mummy?). Stage 1 - verify filled constraints: Does any error occur in filled remaining constraints? No. Go to next stage. Stage 2 - status: Current answer: blank 1: A. Things to Do in Denver When You're Dead. Remaining constraints containing blanks: (blank 2, acted in, Things to Do in Denver When You're Dead); (blank 2, acted in, Scooby-Doo! in Where's My Mummy?). Stage 2 - solve: From (blank 2, acted in, Things to Do in Denver When You're Dead), candidate for blank 2: None of the given candidates satisfies the constraint. There is error in current answer. Go back to previous stage: stage 1. Stage 1 - status: Current answer: None. Remaining constraints containing blanks: (Andy García, acted in, blank 1); (blank 2, acted in, blank 1); (blank 2, acted in, Scooby-Doo! in Where's My Mummy?). Stage 1 - solve: From (Andy García, acted in, blank 1), candidate for blank 1: B. Smokin' Aces. Stage 1 - status update: Current answer: blank 1: B. Smokin' Aces. Filled remaining constraints with current answer: (Andy García, acted in, Smokin' Aces). Updated remaining constraints containing blanks: (blank 2, acted in, Smokin' Aces); (blank 2, acted in, Scooby-Doo! in Where's My Mummy?). Stage 1 - verify filled constraints: Does any error occur in filled remaining constraints? No. Go to next stage. Stage 2 - status: Current answer: blank 1: B. Smokin' Aces. Remaining constraints containing blanks: (blank 2, acted in, Smokin' Aces); (blank 2, acted in, Scooby-Doo! in Where's My Mummy?). Stage 2 - solve: From (blank 2, acted in, Smokin' Aces), candidate for blank 2: C. Jeremy Piven. Stage 2 - status update: Current answer: blank 1: B. Smokin' Aces, blank 2: C. Jeremy Piven. Filled remaining constraints with current answer: (Jeremy Piven, acted in, Smokin' Aces); (Jeremy Piven, acted in, Scooby-Doo! in Where's My Mummy?). Remaining constraints with blanks: None. Stage 2 - verify filled constraints: Does any error occur in filled constraints? No. Go to next stage. Stage 3 - status: Current answer: blank 1: B. Smokin' Aces, blank 2: C. Jeremy Piven. Remaining constraints with blanks: None. Stage 3 - solve: No more remaining constraints with blank. Final answer: blank 1: B, blank 2: C

Instruction: Pick the correct answer for each blank that satisfies all the given constraints.

Knowledge: (Paz Vega, acted in, The Human Contract); (Joe Roberts, acted in, Our Hospitality); (Paz Vega, acted in, The Six Wives of Henry Lefay); (Paz Vega, acted in, The Spirit (film)); (Charlie Chaplin, directed, A Woman of Paris); (Charlie Chaplin, directed, The Champion (1915 film)); (Charlie Chaplin, directed, Limelight (1952 film)); (Jada Pinkett Smith, directed, The Human Contract); (Johnathon Schaech, acted in, Prom Night (2008 film)); (Joe Roberts, acted in, Our Hospitality); (Brittany Snow, acted in, Prom Night (2008 film)); (Idris Elba, acted in, Prom Night (2008 film)); (Joe Roberts, acted in, The Haunted House (1921 film)); (Idris Elba, acted in, The Human Contract); (Joe Roberts, acted in, Our Hospitality); (William Hopper, acted in, The Bad Seed (1956 film)). (Optional)

Constraints: (Paz Vega, acted in, blank 1); (Jada Pinkett Smith, directed, blank 1); (blank 2, acted in, Prom Night (2008 film)); (blank 2, acted in, blank 1).

Options: blank 1: A. The Human Contract, B. The Spirit (film), C. The Six Wives of Henry Lefay blank 2: A. Johnathon Schaech, B. Idris Elba, C. Brittany Snow

Answer:

VERIFY-ALL (ours)

Instruction: Pick the correct answer for each blank that satisfies all the given constraints.

Knowledge: (Charlton Heston, acted in, True Lies); (Eliza Dushku, acted in, True Lies); (Tom Arnold (actor), acted in, True Lies); (Bill Paxton, acted in, True Lies); (Charlton Heston, acted in, Chiefs (miniseries)); (Stephen Collins, acted in, Chiefs (miniseries)); (Paul Sorvino, acted in, Chiefs (miniseries)); (Danny Glover, acted in, Chiefs (miniseries)). (Optional)

Constraints: (blank 1, acted in, True Lies); (blank 1, acted in, Chiefs (miniseries)).

Options: blank 1: A. Bill Paxton, B. Charlton Heston, C. Paul Sorvino

Answer: Candidate answer: blank 1: A. Bill Paxton; Filled constraints with candidate answer: (Bill Paxton, acted in, True Lies); (Bill Paxton, acted in, Chiefs (miniseries)); Verification: Does error occur in filled constraints with candidate answer? (Bill Paxton, acted in, Chiefs (miniseries)) is incorrect. Candidate answer: blank 1: B. Charlton Heston; Filled constraints with candidate answer: (Charlton Heston, acted in, True Lies); (Charlton Heston, acted in, Chiefs (miniseries)); Verification: Does error occur in filled constraints with candidate answer? No. Therefore, blank 1: A.

Instruction: Pick the correct answer for each blank that satisfies all the given constraints.

Knowledge: (Joe Roberts, acted in, Our Hospitality); (Joe Roberts, acted in, Neighbors (1920 film)); (Taye Diggs, acted in, Rent (film)); (Joe Roberts, acted in, Three Ages); (Bradley Cooper, is married to, Jennifer Esposito); (Taye Diggs, is married to, Idina Menzel); (Charlie Chaplin, is married to, Mildred Harris); (Mary, Queen of Hungary, is married to, Jobst of Moravia); (Idina Menzel, acted in, Enchanted (film)); (Idina Menzel, acted in, Rent (film)); (Joe Roberts, acted in, Neighbors (1920 film)); (Joe Roberts, acted in, Three Ages)f; (Idina Menzel, is married to, Taye Diggs); (Bradley Cooper, is married to, Jennifer Esposito); (Mary, Queen of Hungary, is married to, Jobst of Moravia); (Charlie Chaplin, is married to, Mildred Harris). (Optional)

Constraints: (blank 1, acted in, blank 2); (blank 1, is married to, Idina Menzel); (Idina Menzel, acted in, blank 2); (Idina Menzel, is married to, blank 1).

Options: blank 1: A. Kelly LeBrock, B. Napoleon, C. Taye Diggs blank 2: A. Halloweentown High, B. Magnolia (film), C. Rent (film)

Answer: Candidate answer: blank 1: C. Taye Diggs, blank 2: B. Magnolia (film); Filled constraints with candidate answer: (Taye Diggs, acted in, Magnolia (film)); (Taye Diggs, is married to, Idina Menzel); (Idina Menzel, acted in, Magnolia (film)); (Idina Menzel, is married to, Taye Diggs); Verification: Does error occur in filled constraints with candidate answer? (Taye Diggs, acted in, Magnolia (film)) is incorrect. Candidate answer: blank 1: C. Taye Diggs, blank 2: C. Rent (film); Filled constraints with candidate answer: (Taye Diggs, acted in, Rent (film)); (Taye Diggs, is married to, Idina Menzel); (Idina Menzel, acted in, Rent (film)); (Idina Menzel, is married to, Taye Diggs); Verification: Does error occur in filled constraints with candidate answer? No. Therefore, blank 1: C, blank 2: C.

Instruction: Pick the correct answer for each blank that satisfies all the given constraints.

Knowledge: (Andy García, acted in, Smokin' Aces); (Andy García, acted in, Ocean's Thirteen); (Andy García, acted in, The Untouchables (film)); (Andy García, acted in, The Pink Panther 2); (Jeremy Piven, acted in, Smokin' Aces); (Joe Roberts, acted in, Our Hospitality); (Joe Roberts, acted in, Three Ages); (Joe Roberts, acted in, Neighbors (1920 film)); (Virginia Madsen, acted in, Scooby-Doo! in Where's My Mummy?); (Jeremy Piven, acted in, Scooby-Doo! in Where's My Mummy?); (Mindy Cohn, acted in, Scooby-Doo! in Where's My Mummy?); (Grey DeLisle, acted in, Scooby-Doo! in Where's My Mummy?). (Optional)

Constraints: (Andy García, acted in, blank 1); (blank 2, acted in, blank 1); (blank 2, acted in, Scooby-Doo! in Where's My Mummy?).

Options: blank 1: A. Things to Do in Denver When You're Dead, B. Smokin' Aces, C. Beverly Hills Chihuahua blank 2: A. Ron Perlman, B. Casey Kasem, C. Jeremy Piven

Answer: Candidate answer: blank 1: B. Smokin' Aces, blank 2: C. Jeremy Piven; Filled constraints with candidate answer: (Andy García, acted in, Smokin' Aces); (Jeremy Piven, acted in, Smokin' Aces); (Jeremy Piven, acted in, Scooby-Doo! in Where's My Mummy?); Verification: Does error occur in filled constraints with candidate answer? No. Therefore, blank 1: B, blank 2: C.

Instruction: Pick the correct answer for each blank that satisfies all the given constraints.

Knowledge: (Miriam Margolyes, acted in, Harry Potter (film series)); (David Thewlis, acted in, Harry Potter (film series)); (John Cleese, acted in, Harry Potter (film series)); (Richard Harris, acted in, Harry Potter (film series)); (Joe Roberts, acted in, Three Ages); (Maureen Lipman, acted in, A Little Princess (1986 TV serial)); (Nigel Havers, acted in, A Little Princess (1986 TV serial)); (Miriam Margolyes, acted in, A Little Princess (1986 TV serial)). (Optional)

Constraints: (blank 1, acted in, Harry Potter (film series)); (blank 1, acted in, A Little Princess (1986 TV serial)).

Options: blank 1: A. Miriam Margolyes, B. Maggie Smith, C. Emma Watson

Answer: Candidate answer: blank 1: A. Miriam Margolyes; Filled constraints with candidate answer: (Miriam Margolyes, acted in, Harry Potter (film series)); (Miriam Margolyes, acted in, A Little Princess (1986 TV serial)); Verification: Does error occur in filled constraints with candidate answer? No. Therefore, blank 1: A

Instruction: Pick the correct answer for each blank that satisfies all the given constraints.

Knowledge: (Paz Vega, acted in, The Human Contract); (Joe Roberts, acted in, Our Hospitality); (Paz Vega, acted in, The Six Wives of Henry Lefay); (Paz Vega, acted in, The Spirit (film)); (Charlie Chaplin, directed, A Woman of Paris); (Charlie Chaplin, directed, The Champion (1915 film)); (Charlie Chaplin, directed, Limelight (1952 film)); (Jada Pinkett Smith, directed, The Human Contract); (Johnathon Schaech, acted in, Prom Night (2008 film)); (Joe Roberts, acted in, Our Hospitality); (Brittany Snow, acted in, Prom Night (2008 film)); (Idris Elba, acted in, Prom Night (2008 film)); (Joe Roberts, acted in, The Haunted House (1921 film)); (Idris Elba, acted in, The Human Contract); (Joe Roberts, acted in, Our Hospitality); (William Hopper, acted in, The Bad Seed (1956 film)). (Optional)

Constraints: (Paz Vega, acted in, blank 1); (Jada Pinkett Smith, directed, blank 1); (blank 2, acted in, Prom Night (2008 film)); (blank 2, acted in, blank 1).

Options: blank 1: A. The Human Contract, B. The Spirit (film), C. The Six Wives of Henry Lefay blank 2: A. Johnathon Schaech, B. Idris Elba, C. Brittany Snow

Answer:

