# OpenReview forum: "Knowledge Crosswords: Geometric Reasoning over Structured Knowledge with Large Language Models"
_ICLR.cc/2024/Conference — ICLR 2024 Conference Withdrawn Submission_

### Official Review · Reviewer_3S11 · 2023-10-29

**Soundness:** 2 fair
**Presentation:** 3 good
**Contribution:** 2 fair
**Rating:** 3
**Confidence:** 4

**Summary:**

This paper proposes a multi-blank QA dataset, called KNOWLEDGE CROSSWORDS, to evaluate the reasoning ability of LLMs under scenarios where structural knowledge is organized as an interweaving network. The authors further propose two baselines to augment the ability of LLMs to backtrack and verify structural information. Expensive experiments are conducted to evaluate existing LLM-based methods and the proposed baselines.

**Strengths:**

1. This paper explores a question overlooked by existing works: how well LLMs can reason from structural knowledge across a graph.

2. This paper provides a dataset for the above research question, which will contribute to future works in this area.

3. This paper proposes two baselines and conducts massive experiments to evaluate existing LLM-based models and the baselines.

**Weaknesses:**

1. The proposed benchmark seems to be synthetic and less practical. There are natural language questions in traditional QA tasks, as shown in Figure 1. However, in the proposed benchmark, constraints are provided as triplets, which are not coherent sentences. Besides, there is a background KG for KGQA, which can be leveraged by LLMs. However, the background KG is not accessible in the proposed benchmark.

2. Experiments are not comprehensive enough. There are two dataset variants of the proposed benchmark, i.e., open-book and closed-book. But most analytical experiments are conducted based on only one dataset variant with one or two methods.

**Questions:**

1. In "Relevant Knowledge" of Section 2, it is confusing that "open book" denotes no external knowledge while "closed book" means there is external knowledge.

2. It looks like the benchmark is designed specifically for LLMs. Could existing KBQA methods be evaluated on this benchmark?

3. From Table 2, the performance gaps between "easy" and "medium" for all methods are narrow. Do you think it is better to merge "easy" and "medium" into one?

4. In Section 6, most experiments are conducted under the setting "w/o knowledge" with one or two methods. How about the setting "w knowledge"? Furthermore, the proposed baselines are not involved in some analytical experiments.

---

### Official Review · Reviewer_DY7D · 2023-10-31

**Soundness:** 3 good
**Presentation:** 3 good
**Contribution:** 1 poor
**Rating:** 3
**Confidence:** 4

**Summary:**

Knowledge Crosswords presents a novel dataset for evaluating LLM ability to do geometric reasoning on knowledge graphs. The questions contain KG subgraphs with relationships, entities, and “blank” entities to be filled in by the LLM such that all constraints are met. The LLM has to choose between provided options.  The dataset is constructed from YAGO, filtered to be robust to changing information over time. Questions are generated by generating  a k-hop neighborhood around a center entity, filtering in various ways to reduce the graph, selecting blanks, and then choosing negative samples. Negative samples are chosen using a set of three strategies such that the negative choices have certain similarities to the positive ones.

In addition to the dataset, the authors contribute two additional prompting strategies: Staged Prompting and Verify-All. Staged prompting sequentially solves a constraint, updates status, and then verifies the current solution. Verify-All generates solutions, checks for errors, and then tries to remedy the error to produce a new solution.

**Strengths:**

The generation of the dataset is well defined. Exact details were provided in the appendix but were almost unnecessary to understand the construction.

The Verify-All approach makes a good improvement on performance.

The results are well ablated and analyzed. Results cover different settings (w/ knowledge and w/o) on different categories of difficulty, using different model strengths (gpt 3.5 vs 4). Ablations and analysis includes analyzing under a “None of the above” option, varying number of in context examples, the order options are presented in (interesting that this has an effect), number of options per blank, structure of the constraints (It is interesting that  cycle is the easiest structure to solve), and others.

Originality - Work is original and the dataset is novel. The new prompting strategies are also useful for this problem though it is unclear how different they are from other self refinement techniques.
Quality - The work is high quality. The results seem trustworthy and reproducible (Appendix contains a lot of useful reproduction information).
Clarity - The paper is well written and clear, though some examples earlier in the paper could improve that more.

**Weaknesses:**

1:
Does this dataset actually evaluate geometric reasoning of LLMs? I think the main weakness with this work is that it is not clear that geometric reasoning is what is being tested. For instance, in the “with knowledge” setting. The easy and medium questions are getting high 90’s percents with zero-shot prompting. This leads me to believe that for these questions the model does not need to do geometric reasoning and instead can just guess based on semantic matching. This makes sense given that rule #1 and #2 only ensure that the distractors are in the neighborhood and contain a matching relationship type. Looking at some example questions makes this seem like a fairly easy task.

The hard questions are a bit more challenging and do require a little more reasoning as some constraints will be met by distractors. However, the hard dataset  W/o relevant knowledge is much harder, but not because it requires more geometric reasoning. It is not clear how well YAGO matches reality. For instance in Table 8 we have knowledge containing

(Dick Powell, is married to, Joan Blondell);
as well as
(June Allyson, is married to, Dick Powell);
And one of the constraints is (Dick Powell, is married to, blank 2)

Joan Blondell and Dick Powell divorced in 1944. It would be hard for a LLM which knows that to get the right answer since its knowledge is that Joan Blondell and Dick Powell are not married. (Nevermind the fact that they are also deceased).

This means that only the w/ knowledge setting is useful. Or maybe w/o knowledge but using retrieval from YAGO.

2: What is the real world use of this evaluation?
It is hard to see how this relates to a use case people want to evaluate for. LLMs answer problems in natural language, so that is a use case we want to evaluate. Can LLMs do geometric reasoning to provide answers to user questions? Alternatively, there may be some real-world geometric constraint satisfaction problems that need to be solved and LLMs could be way to do it.   Here is a list of natural Language phrasings of the constraints from the examples on page 19:

Who acted in True Lies and Chiefs (miniseries)?

Who is married to Idina Menzel and what movie did they act in together?

Who acted in a movie with Andy Garcia and also acted in “Scooby Doo! Where’s My Mummy?”  Who acted in both A little Princess and Harry Potter?

Not really sure how to do this one: (Dinah Sheridan, acted in, blank 2); (Dinah Sheridan, is married to, blank 1); (blank 1, acted in, blank 2); (Dinah Sheridan, is married to, blank 3); (blank 3, is married to, Jan Sterling).

Dina Sheridan is married to what co-star in what movie? Who is Dina Sheridan married to who is also married to Jan Sterling?

Which co-actor of Paz Vega in a movie directed by Jada Pickett Smith also acted in Prom Night (2008 film)? And what is the movie they co-acted in by Jada Pickett Smith?

There are some valuable questions here. However, when making a dataset for a language model, effort needs to placed on the language of the questions. Some of these questions don’t make a ton of sense. In addition, answering from 2-3 options is not in line with most use cases where the answer should be generated.

(Note: Part of the issue is that isMarriedTo is in fact time sensitive. If it was “is or was married to”, that would be different.)

3: Why is this not compared to knowledge graph question answering (KGQA) datasets? Existing KGQA datasets test a variety of geometric reasoning abilities in different ways. For instance QALD-10 (and precedents) test the ability to answer questions using a knowledge graph, including questions involving aggregation (counts/sums), comparisons, qualifiers/constraints, and other geometric reasoning patterns. In comparison to those datasets, this seems like more of a constant satisfaction problem than a geometric reasoning one.

**Questions:**

Open-book and closed-book don’t seem to align with standard definitions. Open-book usually means there is a corpus of documents or knowledge base/graph that can be retrieved from. Maybe that is what is meant here but “need to solve knowledge crosswords with only their internal parametric knowledge” does not align with that, and is not what is tested.

The questions themselves could be presented in natural language and it might be less confusing. For instance
Section e: “Which co-actor of Paz Vega in a movie directed by Jada Pickett Smith also acted in Prom Night (2008 film)? And what is the movie they co-acted in by Jada Pickett Smith?”
This could potentially be done using an LLM with annotator validation?

Why does an increase in ICT examples not improve performance? It actually does worse on the easy and medium problems? Seems curious?

What are the distribution of relationship types in the constraints and in the knowledge context? Seems like a lot of "acted in" and "is married to" in the examples.

Why was YAGO chosen? Seems like there are more relations and questions that could be generated using Wikidata.

I think there is value in this direction of research. However, I think in dataset construction, more emphasis needs to be put on “why”. Why would someone use this dataset? What are the real world impacts? Alternatively, a dataset could be used to interrogate a model (instead of a benchmark). How are LLMs able to do geometric reasoning? How do they fail? What kinds of geometric reasoning do we care about and what is missed by existing datasets?  Adhering to a more impactful question would lead to more impactful work.

---

### Official Review · Reviewer_iJEx · 2023-11-01

**Soundness:** 3 good
**Presentation:** 3 good
**Contribution:** 2 fair
**Rating:** 5
**Confidence:** 3

**Summary:**

Paper presents a new benchmark called knowledge crosswords to test the abilities of the large language models for complex reasoning tasks. Method used to generate the benchmark along with prompting strategy over LLMs that can solve this benchmarks is presented. Authors show experimental results over the benchmark on different difficulty levels and show the scope for LLMs to improve on hard problems.

**Strengths:**

New benchmark dataset for testing LLMs reasoning abilities beyond single hop and multi hop with constraints and backtracking.
New prompting techniques that help in improving LLM abilities for this benchmark

**Weaknesses:**

Geometric reasoning and geometric constraints are very loosely used in the paper. Not defined properly. I could not get difference between multi-hop complex questions present in KBQA literature to this benchmark. How does some of the complex QA benchmark compare to this benchmark?

**Questions:**

please define what you mean geometric reasoning and geometric constraints formally. I am having difficult time to understand the difference between complex QA to this benchmark data.

---

### Official Review · Reviewer_mbzm · 2023-11-02

**Soundness:** 4 excellent
**Presentation:** 4 excellent
**Contribution:** 2 fair
**Rating:** 6
**Confidence:** 4

**Summary:**

The paper explores if LLM have a geometric reasoning ability to satisfy multiple structural constraints simultaneously.

The authors create a dataset by sampling weakly connected components from a knowledge graph and mask out some entities (at varying levels of hardness), to test if the LLM can fill in the blanks while respecting constraints, in a closed-book (multiple-choice setting). They sample 1:3 positive and negative answers for the multiple-choice. They test different prompting strategies (including two of their proposed ones), with knowledge context sampled from the KG, and without (from the LLMs parametric knowledge alone). They find that LLMs can reason jointly over the constraints, but may not be as robust to "None of the above" options, besides some other ablations.

**Strengths:**

1. Dataset from sampling and masking from encyclopedia KG, with choices that control hardness. Rigorous considerations and explanations of the choices in pre-processing the dataset to the controlled settings of the dataset and experiments. The dataset quality is really tied to the KG knowledge and its completeness and relevance in isolation (so may be useful to test methods in a closed setting that work on this KG, but may not be so useful to generally test LLMs with general world-knowledge and open settings).

2. Experiments, ablation, and presentation is well done in this paper. There is some value in the study, as it examines LLMs (GPT 3.5 and 4) on various levels of hardness in the 1:3 multi-choice setting. It finds that LLMs can reason for multiple constraints.

**Weaknesses:**

1. The dataset is in built in a very controlled setting, but that limits its usefulness in testing LLM's ability for geometric reasoning over multiple constraints. The dataset is derived from YAGO KG, assuming that is the source of ground truth (pre-processed to be definite and unique answers, and removing ambiguities, time and location sensitivity). This may conflict with general knowledge in the LLM, or alternate answers, limiting the evaluation of the LLM's ability for geometric reasoning. (The latter is called out in the limitations of the paper).

2. Closed multiple-choice QA trivializes the problem. Since ~2 blanks on average, each with 3 choices, can be combinatorially solved for. Yet, the LLM doesn't do very well on hard questions, which suggests this seems to be other problems (either the evaluation limitations mentioned above in point 1, or hallucinations due to prompt design, conflicting LLM parametric knowledge), but its not clear from ablations where the gap is.

**Questions:**

1. If the LLM is basically solving a 2 black, 3 options per blank problem, it should be reasonably good at it. If its not, some failure analysis is required to really say where the gap is (as mentioned in weaknesses- pt 2). Is there clear indication of why the LLM cannot solve this for the Hard case.

---

### Author Response · Authors · 2023-11-14

We would like to thank all reviewers for their valuable comments and suggestions. We will continue improving our work and decide to withdraw this work.